# Impacts of reducing scattering and absorbing aerosols on the temporal extent and intensity of South and East Asian summer monsoon

Chenwei Fang[1, 2, *], Jim M. Haywood[2, 3], Ju Liang[2, 4], Ben T. Johnson[3], Ying Chen[5,6], Bin Zhu[1, *]

[1]Key Laboratory of Meteorological Disaster, Ministry of Education (KLME), Joint International Research Laboratory of Climate and Environment Change (ILCEC), Collaborative Innovation Center on Forecast and Evaluation of Meteorological Disasters, Key Laboratory for Aerosol-Cloud-Precipitation of China Meteorological Administration, Nanjing University of Information Science & Technology, Nanjing 210044, China
[2]Faculty of Environment, Science and Economy, University of Exeter, Exeter, UK
[3]Met Office Hadley Centre, Exeter, UK
[4]College of Resources and Environmental Sciences, China Agricultural University, Beijing, 100193, China
[5]Paul Scherrer Institute, Forschungsstrasse 111, 5232 Villigen, Switzerland
[6]School of Geography Earth and Environment Sciences, University of Birmingham, Birmingham B15 2TT, UK

*Correspondence to*: Chenwei Fang (fangcw515@163.com) and Bin Zhu (binzhu@nuist.edu.cn)

**Abstract.** The vast majority of reductions in aerosol emissions are projected to take place in the near future; however, associated impacts on the large-scale circulation over the populated Asian monsoon region remain uncertain. Using the state-of-the-art UK Earth System Model version 1 (UKESM1), this study examines the response of the South Asian and East Asian summer monsoon (SASM and EASM) to idealized reductions in anthropogenic emissions of carbonaceous aerosols and $SO_2$. The analysis focuses on changes in the monsoon temporal extent and intensity of precipitation following decreases in either scattering (SCT), absorbing (ABS) aerosols, or decreases in both. For SCT, the combination of the early transition of land-sea thermal contrast and sea level pressure gradient during the pre-monsoon season together with the late transition in the post-monsoon season associated with the tropospheric warming, advances the monsoon onset but delays its withdrawal, which leads to an extension of the summer rainy season across South and East Asia. The northward shift of the upper-tropospheric Asian jet forced by the SCT reduction causes the anomalous convergence of tropospheric moisture and low-level ascent over northern India and eastern China. The intensification of the South Asian High (SAH) due to the warming over land also contributes to the dynamic instability over Asia. These changes enhance the rainy season of these regions in boreal summer. Reductions in absorbing aerosol act in the opposite sense, making the Asia's rainy season shorter and weaker due to the opposite impacts on land-sea contrast, Asian jet displacement and SAH intensity. With reductions in both SCT and ABS aerosol together the monsoon systems intensify, as the overall impact is dominated by aerosol scattering effects and results in the strengthening of monsoon precipitation and 850-hPa circulation. Although aerosol scattering and absorption play quite different roles in the radiation budget, their effects on the monsoon precipitation seem to add almost linearly. Specifically, the patterns of monsoon-related large-scale responses from reducing both SCT and ABS together are similar to the linear summation of separate effect of reducing SCT or ABS alone, despite of the inherent nonlinearity of the atmospheric systems. The opposing adjustments of

Asian rainy season forced by the ABS and SCT aerosol emission reductions suggest that emission controls that target e.g.

emissions of black carbon that warm the climate would have a different response to those that target overall aerosol emissions.

## 1 Introduction

In addition to anthropogenic greenhouse gas (GHG), it has long been established that aerosol emissions are important external climate forcing that strongly modulates regional climate (e.g. Taylor and Penner, 1994). Previous research has investigated the responses of regional precipitation to the changes in the emissions of aerosols and GHGs, where higher sensitivities are usually

seen for the aerosol-forced responses (Kloster et al., 2009; Liepert et al., 2004; Persad et al., 2022), particularly in regions with high aerosol emissions (Samset et al., 2018). Anthropogenic aerosols can also alter the atmospheric circulation at hemispheric and regional scales due to its inhomogeneous distribution (Cox et al., 1995; Diao et al., 2021). One of the regions where the general circulation is strongly affected by the perturbation of aerosol concentrations is the Asian monsoon region (Jiang et al., 2013; Lau et al., 2017; Li et al., 2018; Salzmann et al., 2014; Undorf et al., 2018). It contributes nearly 20% of global

anthropogenic aerosols emissions (Grandey et al., 2018), covers over 20% of the Earth's land and contains almost 60% of the world's population exposed to monsoon-related climate extremes (Li et al., 2016). Hence, deepening the understanding of the aerosol-related regional climate changes over this region is important for supporting the local communities in coping with possible risks of climate extremes under future changes in external climate forcing.

The South and East Asian summer monsoons (SASM and EASM) are two of the most influential monsoon systems in the

world (Ha et al., 2017). The SASM originates from the northward shift of intertropical convergence zone (ITCZ) and is mainly observed in the tropics, while the EASM has both tropical and subtropical monsoon characteristics (Huang et al., 2017). Previous studies have shown that changes in the anthropogenic aerosol emissions from both local and remote sources can lead to the observed predominant decreased SASM circulation and precipitation during the late twentieth century (Bollasina et al., 2011; 2014). Ganguly et al. (2012) further demonstrated that the fast response of SASM to changes in aerosol emissions

dominates the drying trend in precipitation over the highly populated central-northern Indian region compared with the slow feedbacks associated with aerosol-induced changes in sea surface temperature (SST). However, changes in summer precipitation vary on a regional basis over East Asia with the weakened low-level wind, featuring a decrease in central and northeast China but an increase in the south China and Yangtze River valley in the late 20th century (Li et al., 2016). Anthropogenic aerosols also play a crucial role in the spatial pattern of EASM adjustments (Menon et al., 2002; Rosenfeld et

al., 2007; Jiang et al., 2013; Song et al., 2014; Wang et al., 2017). Interestingly, the weakened wind and accompanying EASM precipitation changes are also modulated by the atmospheric response to aerosol forcings, though the SST cooling also contributes (Wang et al., 2017; Wang et al., 2019). Considering the prominent contributions of the fast atmospheric adjustments to the aerosol-induced changes in both SASM and EASM, this study mainly examines the fast monsoon responses to aerosol emission perturbations.

Globally, emissions of anthropogenic aerosols and their precursors have been declining for the last few decades and the increasing trend of aerosol emissions in many regions of Asia have reversed or are projected to reverse due to technological advances, socioeconomic progress and policy control (Wang et al., 2012; Westervelt et al., 2018). For instance, global emissions of black carbon (BC, absorbing aerosol) and sulfur dioxide ($SO_2$, precursor of scattering aerosol sulfate) in 2100 are projected to decline by 70-90% from 2015 under the medium and strong pollution controls of the Shared Socioeconomic

Pathways (SSPs; Lund et al., 2019). In the Asian monsoon region, $SO_2$ and BC emissions have declined by 75% and 30% respectively over China in the last decade (Li et al., 2017; Zheng et al., 2018). Sulfate and BC burdens over South Asia in 2100 are also assumed to decrease by 11-69% and 0.6-88% respectively compared to those at present day under high- to low-emission scenarios (SSP3-7.0, SSP2-4.5 and SSP1-1.9; Lund et al., 2019). Many studies based on multi-models have considered the global and regional climate adjustment under different levels of air pollution control (e.g., Kloster et al. 2009;

Wang, Z. et al., 2016; Xing et al., 2016; Westervelt et al. 2018; Wilcox et al., 2020). However, the potential changes of EASM and SASM caused by anthropogenic aerosol reductions and the corresponding causal mechanisms have received less attention. In particular, although the physical impacts of scattering (SCT; e.g., sulfate) or absorbing (ABS; e.g., BC) aerosol types are very different (e.g. Li, J. et al., 2022), the impacts of reducing sulfate precursor emissions relative to BC in possible Asian monsoon adjustments remains unclear. Using Community Earth System Model, Zhao et al. (2018) showed a warmer and wetter

Asian monsoon region with increased extreme precipitation events under Representative Concentration Pathway 8.5 scenario but did not focus on the EASM and SASM responses to the change of individual aerosol species.

Besides, most studies focusing on the modulations of EASM and SASM by anthropogenic aerosols only involved the seasonal climate adjustments in summer or winter. Bollasina et al. (2013) pointed out that the anthropogenic aerosols can also lead to sub-seasonal changes to the activity of monsoon systems, i.e., the onset of SASM in spring is observed to become earlier

during the late 20th century and this change is attributed to the increases in the emissions of anthropogenic aerosols. Wang, D. et al. (2016) also reported that the direct effects of BC or the combination of BC and sulfate can bring forward EASM onset due to the deep heating in the middle-upper troposphere in spring. However, it remains unclear how the reductions of SCT and ABS aerosols affect the onset and withdrawal dates of the EASM and SASM. Whether the temporal extent of EASM and SASM will be prolonged or shortened due to the aerosol emission reductions is still unclear.

Considering the difference in the mixing states of aerosol types due to the varying regions and environments, the monsoon responses to the total anthropogenic aerosol forcings may not be a simple linear superposition of the forcings from ABS and SCT aerosols (Ji et al., 2011). Herbert et al. (2021) found that the response of Asian summer monsoon to simultaneous aerosol emission reductions in both South and East Asia differs from the sum of responses to the aerosol reduction in each subregion according to the simulation from an Intermediate Global Circulation Model, implying the nonlinear response of Asian summer

monsoon to regional reductions of aerosol emissions. However, the responses of the Asian summer monsoon to the reduction of total anthropogenic aerosols and different aerosol types and their potential linear combination remain unexamined. Also, further investigations are still required to understand which aerosol types plays the dominant role in shaping the response of EASM and SASM to anthropogenic aerosol reductions.

Impact by the COVID pandemic, a significant global reduction in aerosol and GHGs emissions has been observed in recent
years (Lal et al., 2020; Liu et al., 2020), which appears to affect global and regional climates (Le et al., 2020; Li et al., 2020;
Yang et al., 2020), inspiring us to assess the potential climate adjustments to a temporary short-term perturbation of
anthropogenic aerosol emissions. In a recent study by Fahrenbach and Bollasina (2022), the strong contribution from abrupt
and rapid near-term aerosol emission changes to large-scale climate adjustments at short timescales (i.e. monthly) via rapid
circulation adjustments was highlighted. The signal of this aerosol-induced hemispheric-wide climate adjustments on short
timescales are even consistent with longer-term (decadal) trends according to their study. Hence, this study examines and
compares the responses of the EASM and SASM patterns to the short-term emission reductions of total anthropogenic aerosols
as well as the SCT and ABS types using the state-of-the-art UK Earth System Model version 1 (UKESM1). The temporal and
spatial responses of the large-scale monsoon systems and the environmental mechanism behind these responses are
investigated. Furthermore, the linear relationships between the impacts of emission reductions in different aerosol types will
be discussed. The experiment design, configurations of the UKESM1 model and definitions for monsoon subseasonal
transition are described in Section 2. Section 3 presents results of model verification and experiments quantifying the respective
impacts of reducing total aerosols and different aerosol types on the temporal extent and intensity of SASM and EASM. Section
4 summarize and discuss the findings of this study.

## 2 Methods

### 2.1 Model

This study uses the UKESM1 as described by Sellar et al. (2019). The modelling system is built on top of the core physical
model HadGEM3-GC3.1 (Hadley Centre Global Environment Model version 3; Kuhlbrodt et al., 2018; Williams et al., 2018)
and interactively coupled with the atmospheric components UKCA model (U.K. Chemistry and Aerosols model; Archibald et
al., 2020; Mulcahy et al., 2018), terrestrial biogeochemistry and ocean biogeochemistry. As a successor to HadGEM2-ES
(Hadley Centre Earth System Model; Collins et al., 2011), UKESM1 introduces an improved representation of aerosol radiative
forcing with a prognostic aerosol scheme interacting with radiation and cloud microphysics and a prognostic atmospheric
chemistry scheme, allowing improved representation of the climate responses to short-lived climate forcers (e.g., methane,
tropospheric ozone and BC) reductions (Stohl et al., 2015). The model's resolution is 1.25° in latitude and 1.875° in longitude
and with 85 hybrid height layers from surface to 85 km.

The interactive aerosols are simulated with GLOMAP-Mode (Global Model of Aerosol Processes; Mann et al., 2010), a
double-moment modal aerosol microphysics scheme that represents sulphate, BC, organic carbon (OC) and sea salt across five
log-normal size modes, while mineral dust is simulated with the bin emission scheme of Woodward (2001). Atmospheric
chemistry is simulated using the unified stratospheric-tropospheric (StratTrop) chemistry, which combines the stratospheric
scheme (Morgenstern et al., 2009; 2017) and the tropospheric chemistry (O'Connor et al., 2014). Aerosol radiative effects in
both the shortwave and longwave spectral regions are included (Bellouin et al., 2013). Aerosol-cloud interactions are simulated

using the UKCA-Activate scheme (West et al., 2014). For the simulations used here, the prescribed GHG concentrations (including CFC-12, $CH_4$, $CO_2$, HFC-134 and $N_2O$, etc.) are from the Coupled Model Project Intercomparison Phase 6 (CMIP6; Meinshausen et al., 2020) under the SSP3-7.0 scenario; for more details see O'Connor et al. (2021).

## 2.2 Experimental Design

Four sets of simulations were conducted in this study. Each comprised a 10-member ensemble of UKESM1 simulations that ran for 5 years from 2020 to 2024. In the control set all forcings and aerosol emissions were based on the CMIP6 SSP3-7.0 scenario (Rao et al., 2017), which arguably represents one of the possible "worst" future pathways for air quality with weak pollution controls and low levels of technology development. Lund et al. (2019) estimated that the total aerosol forcing in 2100 relative to 1750 is -0.51 W m$^{-2}$ in SSP3-7.0, which is similar to estimates of the preindustrial to present-day level (IPCC, 2021).

However, the simulations in this study stop in 2024, when aerosol emissions in SSP3-7.0 are still close to those in other SSP scenarios according to the estimation of Lund et al. (2019). Hence, the control simulations based on SSP3-7.0 scenario give a reasonable baseline prediction of the period assuming typical levels of emissions persist.

To investigate the theoretical impacts that short-term pollution mitigation may have, three other sets of simulations were performed in which aerosol emissions were perturbed globally in different ways. In the "Total" set, all anthropogenic emissions

of $SO_2$, organic matter (OM) and BC were reduced by 75% relative to the SSP3-7.0 scenario. This included perturbing emissions from all fossil fuel and biofuel sources but not from biomass burning. In the "SCT" set only the $SO_2$ and OM emissions were reduced (again by 75% relative to SSP3-7.0) to decrease the loading of sulphate and organic aerosols that predominantly scatter solar radiation. In the "ABS" set only the BC was reduced (by 75% relative to SSP3-7.0), which substantially decreases the absorption of solar radiation by aerosol. BC was the only anthropogenic aerosol component in the

model to absorb significantly in the solar spectrum. The aim of selectively reducing aerosol scattering and absorption was to understand the role of these different aerosol-radiative interactions on the monsoons, whereas the Total experiment allowed us to investigate if reducing scattering and absorbing together gives a different response compared to summing effects from reducing them separately. The Asian monsoon adjustments forced by pollution mitigation are diagnosed as the difference between the aerosol-emission-perturbed and control runs.

In each of the perturbed sets the aerosol reductions were applied globally but only for the first 2 years of the simulations, after which emissions returned to the levels set by the SSP3-7.0 scenario. Only the first 2 years of the perturbed simulations were used in this study, whereas the data from the final 3 years offer opportunities for follow-on studies to look at the response to suddenly withdrawing pollution controls. To create the 10 member ensembles within each set the individual simulations ran with different random perturbations in the stochastic physics, causing the atmospheric flow to diverge into different

meteorological realizations. The stochastic physics introduces small random perturbations to the wind fields, temperature and moisture tendencies from some of the sub-grid parameterizations schemes including convection, gravity-wave drag, radiation and large-scale cloud microphysics. These perturbations are applied in a way that conserves energy, momentum and moisture but represents variability and uncertainty in unresolved physical processes, which has been shown to improve ensemble

predictions on medium-range (Palmer et al., 2009; Tennant et al., 2011), seasonal (Weisheimer et al., 2011) and decadal timescales (Doblas-Reyes et al., 2009). Using the first 2 years of each simulation gives us the equivalent of a 20 years sample from each set, allowing a statistically more significant picture of the climate in the control case and those with the aerosol emissions perturbations.

It should be noted that the model includes aerosol-radiation interactions, aerosol-cloud interactions and surface albedo effects, but this study will mainly discuss aerosol-radiation interactions. To evaluate the performance of UKESM1 in simulating the Asian summer monsoon we also use precipitation and wind fields from the period 1985-2014, obtained from the UKESM1 historical simulations in the CMIP6 database (https://esgf-node.llnl.gov/search/cmip6/). The basic model configuration and resolution of the UKESM1 historical simulations are consistent with the simulation settings used in this study except for aerosol emissions and time-varying forcings. To evaluate precipitation over the SASM and EASM regions we also use observational precipitation data from both the Climate Prediction Center (CPC) unified gauge-based daily observations (Chen et al. 2008) and the Global Precipitation Climatology Project (GPCP) rain gauge-satellite combined precipitation dataset (Huffman and Bolvin, 2013). Precipitation and wind fields from the ECMWF's (European Center for Medium-Range Weather Forecast) Fifth-generation Reanalysis (ERA5; Hersbach et al., 2020) are also used in comparisons against the model simulations.

**2.3 Definitions for monsoon onset and withdrawal**

Monsoon transition is usually referred to as the seasonal shift of wind direction between the dry and wet seasons (Zhao et al., 2006). The change of some key climatic variables in the monsoon region is often used to define the onset and withdrawal pentad (5-day mean) or onset and withdrawal day for both the SASM and EASM (e.g., He and Zhu, 2015; Noska and Misra, 2016; Wang, D. et al., 2016). Note that the SASM and the continental part of the EASM are regarded as tropical and subtropical monsoons, respectively, and their seasonal wind reversals are mainly characterized by the changes of zonal and meridional winds (Sun and Ding, 2011). In this study, the monsoon duration and the precipitation for the duration is obtained by calculating the monsoon onset and withdrawal dates. The monsoon onset/withdrawal dates are derived according to the definitions given in previous studies. The monsoon changes were calculated based on different definitions as there are significant variations in these parameters under different definitions.

The definitions from Wang et al. (2009) and Noska and Misra (2016), hereafter referred to as W2009 and N2016, are adopted to obtain the SASM onset and withdrawal dates. W2009 uses 850-hPa zonal wind averaged over South Asia (5-15°N, 40-80°E) as an onset circulation index (OCI) of the SASM, and the date of onset is defined as the first day when OCI exceeds 6.2 m s$^{-1}$. N2016 uses All-India rainfall (AIR) to calculate the cumulative pentad mean anomaly $C'_m(i)$ of AIR for pentad $i$ of year $m$: $C'_m(i) = \sum_{n=1}^{i}[AIR_m(n) - \bar{\bar{C}}]$, where $\bar{\bar{C}}$ is the climatology of the annual mean of AIR over N (=72 based on UKESM1's calendar) pentads for $M$ (=2) years. The onset/withdrawal of SASM is defined as the day after $C'_m(i)$ reaches its absolute minimum/maximum. Definitions from Wang, D. et al (2016) and Guo (1983), hereafter referred to as W2016 and G1983, are applied to calculate the EASM monsoon duration and precipitation. The 850-hPa meridional wind ($V_{850}$) over East Asia was

used in W2016 to determine the EASM onset and withdrawal: (1) the onset pentad of the EASM is the pentad when $V_{850}$ over East Asia starts to be greater than 0 m s$^{-1}$ (i.e. a net southerly component) and remains positive in the subsequent three pentads (or the average $V_{850}$ of the accumulative four pentads is greater than 0.5 m s$^{-1}$) ; (2) the withdrawal pentad of the EASM is the

pentad when $V_{850}$ turns negative (i.e. a net northerly component). The EASM onset/retreat pentad based on G1983 was calculated as the difference between the sea level pressures over land (represented by 110°E) and sea (represented by 160°E) over East Asia.

## 3 Results

### 3.1 Model evaluation

The performance of UKESM1 in representing the tropospheric environment has been verified with respect to climate observations, including the variations in aerosol optical depth and atmospheric fields (including temperature, sea level pressure, precipitation and wind fields) on global and regional scales (Mulcahy et al., 2018; Archibald et al., 2020; O'Connor et al., 2021). As our focus is primarily on the monsoon, here the model performance is evaluated by comparing against observations of the regional precipitation over South and East Asia. The division of South and East Asia (Fig. S1) used in this study follows

Iturbide et al. (2020), which is adopted by IPCC (2021). As our study involves the sub-seasonal variations in monsoon onset and withdrawal, the monthly comparisons among South and East Asia precipitations from CPC and GPCP observations, ERA5 reanalysis and the UKESM1 simulations are shown in Fig. 1. The mean values of observations from the GPCP and CPC datasets are shown as black dots. Simulated precipitation over South Asia from the UKESM1 model shows higher correlations (0.69-0.81) from July to September with observations than that (0.54-0.58) from May to June, indicating the better performance

of UKESM1 in reproducing the summer and early autumn precipitation, although all correlations are statistically significant ($p < 0.001$). The precipitation variations over East Asia are also effectively reproduced by the model with correlations of 0.63-0.86 ($p < 0.001$). The standard deviation (STD) reflects the variation range of the dataset. A normalized STD (STD$_{model}$/STD$_{observations}$) value of 1 indicates the same varied amplitude between the simulations and the observations. The normalized STD values range from 0.98-1.49 over South Asia and 1.06-1.38 over East Asia from May to September, which is

within the range of 0.4-2.0 calculated from the simulated climatological mean summer precipitation based on CMIP5 and CMIP6 climate models in Xin et al. (2020) who performed comparisons over East Asia. The root-mean-square deviation (RMSD) indicates the departure between the simulations and observations. Khadka et al. (2021) showed that the RMSE ranges based on CMIP5 and CMIP6 models are 2.18-4.01 and 1.91-6.0 mm day$^{-1}$, respectively, over Southeast Asian monsoon region. The RMSE range between the UKESM1 simulation and observation is 3.04 - 5.35 mm day$^{-1}$ and 1.37 - 2.26 mm day$^{-1}$ over

South and East Asia, respectively, which is within the RMSE range of CMIP6 models.

Fig. 2 shows the wind fields over Asia continent during pre-monsoon (April-May), monsoon (June-August) and post-monsoon seasons (September-October). The selection of the different monsoonal periods follows previous studies (Vissa et al., 2013; Zeng et al., 2019; Zhou et al., 2020). Overall, the model captures the spatial features and temporal evolutions of the ASM

horizontal circulations from upper to lower levels. However, it should be noted that the simulated upper-level westerly jet
shows a positive difference northward of 40°N and a negative difference around 30°N from pre- to post-monsoon seasons
compared to ERA-5 reanalysis (Fig. 2g-i), indicating a slightly wider but less intensive westerly jet in the UKESM1 simulation,
especially for the pre-monsoon season. The lower-level southwest monsoon flow over South Asia is also overestimated in the
model, while the monsoon southerly wind prevailing over East Asia between 20 and 40°N is slightly underestimated (Fig. 2q).

## 3.2 Response of monsoon temporal extent and intensity

Fig. 3 shows the variations in monsoon duration and monsoon precipitation of SASM and EASM induced by the modelled
aerosol reductions, and the quantitative results are summarized in Table S1.The SASM duration and precipitation show similar
changes in Fig. 3(a) and (b) but have different variations in range in W2009 and N2016. Generally, reduction in SCT extends
the temporal extent of the SASM duration and enhances the monsoon precipitation compared to the SASM in the control case,
while reduction in ABS shortens the SASM and reduces the monsoon precipitation. With the combined effects induced by
SCT and ABS aerosols, the SASM temporal extent of precipitation responses induced by the reduction in total aerosols follows
the impacts of the SCT reduction, although the enhancement is weaker than pure SCT reduction. To determine the SASM
duration changes, the variations in monsoon onset and withdrawal dates are further examined (Fig. 4). Reduction in SCT
advances the SASM onset but delays the SASM withdrawal, thus extending the SASM duration to a certain extent (0.4 pentads
in W2009 and 2.4 pentads in N2016). However, reduction in ABS tend to delay the SASM onset (more pronounced in W2009)
and advance the SASM withdrawal, thus shortening the SASM duration by 2 pentads (W2009) or 1 pentad (N2016).
Simultaneously influenced by the SCT and ABS reductions, reduction in total aerosols shows limited impacts on the SASM
onset and withdrawal (advances the SASM onset in N2016).

Note that using different definitions of monsoon onset/withdrawal dates may result in the variations in the monsoon duration
response range although the SASM duration and precipitation adjustments in W2009 and N2016 are qualitatively consistent.
The SASM durations in N2016 from different simulation sets are basically 4-5 pentads longer than those in W2009 (Fig. 3a
and b). The SCT-driven extension of the SASM duration based on N2016 (2 pentads) is also longer than that in W2009 (0.4
pentads). The difference in the SASM duration adjustments between W2009 and N2016 can be attributed to the distinct
selection of monsoon feature to characterize the monsoon subseasonal variations. Syroka et al (2004) pointed out that the
withdrawal of the SASM defined by the precipitation is much later than that defined by the monsoon circulation due to the late
decrease in precipitation in southern India. The precipitation continues to increase in southern Indian after September
associated with the winter monsoon (Bhanu Kumar et al., 2004), while the SASM-related circulation characteristics becomes
unclear in the meantime. Therefore, the SASM onset dates based on N2016 is roughly the same with those based on W2019,
but the withdrawal date is about 5 pentads later, resulting in the longer monsoon duration (Fig. 4a and b). Moreover, there
exists an additional enhancement of monsoon precipitation over SA in the "SCT" set, which further leads to the later SASM
withdrawal and longer SASM duration in N2016 (Fig. 4b). Besides, the precipitation during early autumn is sensitive to the
location and synoptic/sub-synoptic systems (tropical cyclones, depressions, easterly waves, north-south trough activity and

coastal convergence, etc; Bhanu Kumar et al., 2004), which possibly contributes to the larger variation range in the monsoon withdrawal date in N2016.

The impacts of reducing SCT and ABS on the EASM in terms of timescale and intensity (here is characterized by precipitation amount) are similar to that on the SASM, except that the reduction in total aerosols slightly shortens the temporal extent of the EASM (more pronounced in G1983) and increases the summer precipitation over the EASM-controlled region (Fig. 3c and d). The monsoon duration is extended by about 1 pentad both in W2016 and G1983 due to the reduction in SCT, which is mainly from the monsoon withdrawal deferment (Fig. 4c and d). Reduction in ABS oppositely advances the withdrawal, leading to a shorter monsoon (1 pentad in G1983) in East Asia. Compared to the distinguishable EASM withdrawal adjustments, the SCT- or ABS-reduction induced EASM onset adjustments calculated by W2016 (based on meridional wind) and G1983 (based on land-sea pressure difference) are not obvious and consistent, indicating the complexity of EASM onset. He et al. (2008) also pointed out that the EASM exhibits a progressive and complicated establishment and a swift withdrawal. The EASM onset date is postponed but the withdrawal date is advanced due to the total aerosol reduction, hence the EASM temporal extent is shortened a little (0.5 pentads in W2016 and 1.4 pentads in G1983). Compared to the EASM adjustments in W2016, the EASM show longer duration (about 3 pentads) in G1983 due to the later withdrawal (about 4 pentads). Zhu et al. (2012) has clarified that the climatological transition date of the zonal land-sea contrast in autumn over EASM-controlled region is about 3 pentads later than that of the monsoon circulation, which largely explained the relatively late monsoon withdrawal dates in G1983.

Fig. 5 and 6 show the spatial patterns of the opposing changes in the precipitation and the 850-hPa circulation of the SASM and EASM induced by the SCT and ABS reductions. The monsoon precipitation changes over SA are consistent in W2009 and N2016, showing significantly increased (decreased) precipitation due to SCT (ABS) reduction. The low-level SASM circulation is also enhanced (weakened) over the Indian peninsular with the reduced SCT (ABS) based on W2009, while the wind response is different in N2016. The wind field adjustment in N2016 is characterized by a weak southwesterly anomaly over the north-central part of the Arabian Sea (north of 20°N) but an easterly anomaly over the south India and south Arabian Sea (10-20°N). The enhancement of easterly flow over SA could be associated with the relatively late monsoon withdrawal dates (58th pentad; Table S1) based on N2016 in the "SCT" set. The continuously increasing precipitation related to winter monsoon in the southern part of SA after September (Syroka et. al, 2004) and the SCT-reduction-induced increased precipitation in SA (Fig. 4g) jointly lead to the delay of the SASM withdrawal date to October based on N2016. At this time, the low-level circulation over south SA and south Arabian Sea is dominated by the prevailing easterly (October-December; Sengupta and Nigam, 2019) although the local precipitation remains elevated, and is associated with the summer monsoon precipitation according to the N2016. Hence, the onset is better defined than the withdrawal based on the precipitation definition adopted in N2016, especially over southern SA. The W2009 definition is more widely applicable over SA, and the summer monsoon precipitation increase is more logically coherent with the circulation enhancement based on this definition. The anomalies of monsoon precipitation and circulation for EASM are more complicated. Accompanied by the abundant moisture brought by the stronger southwesterlies from Bay of Bengal and Indian Ocean, reduction in SCT increases the precipitation over most parts of eastern China and enhances the EASM circulation with enhanced southwesterly and

southeasterly 850-hPa wind anomalies, especially over the southern part. Reduction in ABS mainly induces a decrease of precipitation in different subregions over East Asia, but the decrease is significant only in the regions with large changes. There are also some regions with increased anomalous precipitation, but most of the precipitation increase was not statistically significant in this study ($p < 0.05$). Besides, the EASM circulation adjustments related to the ABS reduction enhances the northwesterly wind anomalies over North China, Korean Peninsula and Japan, thus weakening the EASM circulation to some extent. Overall, the EASM adjustments in terms of the temporal extent and intensity calculated based on G1983 are basically consistent with the results based on the W2016, in spite of the relatively late monsoon withdrawal dates in G1983, which adopts the land-sea pressure difference as the key monsoon characteristic.

In general, the impacts of the SCT reductions dominate both the SASM and EASM adjustments related to the monsoon precipitation increase and circulation enhancement induced by short-term total aerosols mitigation. It should be noted that the SCT reduction only dominate the precipitation increase over the north-eastern SA (north of 22°N) induced by the total aerosol reduction. The precipitation decrease over central SA (south of 22°N) is contributed by the impacts of ABS reduction and non-linear effects between the SCT and ABS, but the decrease is insignificant in both W2009 and N2016 (Fig. 5).

Future global emission reductions of the SCT and ABS aerosols may not be synchronous due to the differences in contributing region and sector sources, technological progress and air pollution policies (Li, H. et al., 2022; Rao et al., 2017). However, the SASM and EASM responses to the reductions in total aerosols may not be a linear summation of the impacts of the reductions in individual aerosol type due to the nonlinearity of the atmospheric systems. Therefore, we compare the results summed from the sensitivity experiments of reducing SCT or ABS alone with those of reducing both of them simultaneously to estimate the importance of the nonlinear atmospheric adjustments on the monsoon changes in the future and investigate the respective theoretical impacts of simultaneous or non-simultaneous emission reductions of the SCT and ABS aerosols on the Asian region. Generally, the pattern of the anomalous precipitation and monsoon horizontal circulation over SA by adding the results of reducing SCT and ABS aerosols are similar to the results of reducing total aerosols, especially for the W2009 (Fig. 5). However, the precipitation north of 30°N shows a reduction in the validity of the linear addition assumption compared to the precipitation change in the simulation of reducing total aerosols, although most of the reduced precipitation does not pass the significance test ($p < 0.05$). There's also significantly increased precipitation over the southern part of SA in the linear addition, which is contributed by the impacts of SCT reduction. Additionally, an easterly anomaly appears over the Arabian Sea (10-20°N) in N2016 (Fig. 5f) as the linear addition of Fig. 5g and 5h, while an SASM westerly flow is enhanced over this region in the simulations of reducing total aerosols (Fig. 5e). The dominated impacts of SCT reduction and non-linear effects between the SCT and ABS contribute to the enhanced SASM westerly in Fig. 5e. The general feature of precipitation and circulation responses over the EA continent (north of 15°N) in the linear addition are also consistent with that in the simulation of simultaneous SCT and ABS reductions (Fig. 6), except for the insignificant decreased precipitation contributed by the impacts of ABS reduction (Fig. 6b and 6f). For the quantitative results of regional precipitation adjustments, the linear addition results show an increased precipitation in both SA and EA compared with the CTRL results (Fig. 3). The increased precipitation amount is less than the results of reducing total aerosols due to the simple addition of precipitation change caused by ABS

reduction. However, the results of linear addition are inconsistent with the total aerosol reduction results in terms of the SASM and EASM duration variations, indicating that the impacts of reducing SCT or ABS alone on monsoon subseasonal variability cannot be simply added up.

## 3.3 Responses of the monsoon-related large-scale environments

### 3.3.1 Responses of land-sea contrast

We now examine how the atmospheric adjustments regulate the duration and intensity of the SASM and EASM due to the aerosol reductions. Figure 7 shows the time-altitude cross sections of air temperature anomalies for emission reductions in different aerosol types averaged over South and East Asia. The weakened aerosol scattering significantly increases the tropospheric temperature over South and East Asia land throughout the year, while the ABS reduction leads to a decrease in air temperature in the tropospheric due to the weakened absorption of shortwave radiation. This air temperature increase

(decrease) due to the SCT (ABS) reduction over Asia during pre- and post-monsoon seasons is favourable for early (late) transition of land-sea thermal contrast in spring and late (early) transition in autumn, provoking the early (late) monsoon onset and late (early) withdrawal (Wang, D. et al., 2016).

The adjusted land-sea thermal contrast contributes to the significant sea level pressure (SLP) anomalies over the Asian continent and the adjacent oceans during pre-monsoon, monsoon and post-monsoon seasons (Fig. 8). The SCT reduction

induced land warming reduces the SLP over Asia continent but increase the SLP over Northwest Pacific (Fig. 8c, g and k). The quantitative results of the anomalous land-sea SLP difference between the Asian continent and its surrounding oceans and seas are shown in Fig. S3. The SCT reduction induces a negative land-sea SLP difference anomaly throughout the year (Fig. S3 and Fig. 8c, g and k), which is favourable for the advance in the land-sea SLP difference transition from positive to negative in spring and the delay in the transition from negative to positive in autumn. The negative anomalous land-sea SLP difference

also leads to bigger land-sea SLP contrast and a stronger SASM and EASM circulation in the monsoon season. Note that the SLP changes in part of the oceanic areas adjacent to the SA region are consistent with the continental SLP changes, albeit with a smaller range of decrease (Fig. 8c, g and k). This could potentially be attributed to the reduced ACT that transported from Asian continent. However, the SLP decrease over these oceanic areas exerts negligible influence on the overall SCT-reduction-induced anomalous negative land-sea SLP difference between the Asian continent and adjacent oceans (Fig. S3). In contrast,

the ABS reduction leads to a SLP increase over Asian land and part of its surrounding oceans (Fig. 8d and h), resulting a positive land-sea SLP difference anomaly during pre-monsoon and monsoon seasons (Fig. S3); hence, the land-sea SLP difference reversal is delayed in pre-monsoon season, thus shortening the duration of both the SASM and EASM. The land-sea SLP contrast is reduced during monsoon season and weakens the Asian monsoon intensities because of the positive land-sea SLP difference anomaly.

In addition, the anomalous land-sea SLP difference between the Asian continent and the topical Indian and Northwest Pacific Oceans caused by the short-term total aerosols mitigation during monsoon season is dominated by the SCT aerosols and

enhances the monsoon circulation over South and East Asia (Fig. 8e). There is also a negative land-sea SLP difference anomaly due to the total aerosols mitigation in pre- and post-monsoon seasons (Fig. S3b), which is governed by the impacts of SCT-reduction. However, the spatial pattern of SLP adjustments during pre-monsoon season induced by total aerosol reduction

shows a SLP increase over the seas of Southeast Asia, and both the impacts of SCT- and ABS-reduction (Fig. 8 c and d) contribute to the SLP increase over this region. Besides, the ABS-reduction has no significant impacts on the SLP adjustments during post-monsoon season (Fig. 8i). But the regions with significant SLP changes caused by total aerosol reduction are also inconsistent with those caused by the SCT reduction (Fig. 8i and k), indicating the strong non-linearity of atmospheric system. For the adjustments of air temperature (Fig. 7) and land-sea SLP difference (Fig. 8) in the linear addition, their general features

are also coherent with the results of reducing the total aerosols, yet there exist differences in details. For example, there is a significant SLP increase in the Northwestern Pacific Ocean in Fig. 8f due to the simple addition of the impacts of SCT and ABS reductions from Fig. 8g and 8h while reduction in total aerosols induces insignificant SLP changes over this region (Fig. 8e). Nonetheless, both results of linear addition and reducing total aerosols yields negative anomalies of land-sea SLP difference during monsoon season.

### 3.3.2 Responses of the upper-tropospheric systems

For the upper troposphere, we first analyze the responses of the subtropical westerly jet (SWJ) and tropical easterly jet (TEJ) to the reduction in different types of aerosols as these systems are recognized as one of the major circulation systems controlling the Asia climate. Numerous studies have suggested that the summer drought and flood in Asia are closely related to the location and intensity of the upper-tropospheric jet (e.g., Chiang et al., 2017; Madhu, 2014; Xie et al., 2015). However, CMIP6 analysis

from Dong et al. (2022) showed that the anthropogenic aerosols were likely the primary driver of summer jet adjustments. The responses of the South Asian High (SAH) governing the Tibetan Plateau during boreal summer is also considered. The strong association between the variations in the SAH and Asia precipitation anomalies have been proved based on reanalysis and observations (Cai et al., 2017; Wang, L. et al., 2016; Wei et al., 2015; 2017; 2021).

Fig. 9 shows the responses of zonal-mean geopotential height over South (70-90°E) and East Asia (100-120°E) during

monsoon season according to the definitions of W2009 and W2016. Coherent with the temperature perturbations shown in Fig. 7, both the geopotential height changes over South and East Asia during monsoon season caused by the emission reduction of total aerosols are dominated by the atmospheric warming associated with the SCT reduction. The geopotential height increases in the uppermost troposphere (200-500 hPa) around 40°N over South and East Asia, which strengthens (weakens) the poleward pressure gradient force in the north (south) flank of this area. The SWJ axis is centered around 40°N in summer

with central values of 25-30 m s$^{-1}$ at 200-hPa level (Yu et al., 2021), as shown in Fig, 10 and Fig. 11 (a). Therefore, the strengthened pressure gradient force north of the SWJ center induced by the total aerosol/SCT reduction accelerates the westerlies in the north flank of the jet center (north of 40°N over South Asia and north of 50°N over East Asia), and thus the SWJ moves northward over Asia. As can be seen in Fig. 11(a), the strength of TEJ (centered between 10°N and 20°N) is much weaker than that of SWJ. The weakened poleward pressure gradient south of 40°N leads to a negative zonal wind anomalies

over South and East Asia when reducing total or SCT aerosols, leading to the northward shift of the TEJ over South Asia. On the contrast, reduction in ABS decreases the geopotential height in the uppermost troposphere around 40°N and weakens (strengthens) the poleward pressure gradient force in the north (south) flank of the SWJ center, inducing the accelerations of the westerlies in the south flank of the jet center and the southward displacement of the Asian jet over South and East Asia. The geopotential height and upper-tropospheric jet changes in monsoon season based on other definitions (N2016 and G1983)

are shown in Fig. S4 and S5, which have consistent meridional variations with those presented in Fig. 9 and 10.

The meridional shift of the Asian jet and the variations in 200-hPa horizontal circulation field caused by the emission reductions in different aerosol types in monsoon season can be seen more clearly in Fig. 11. The SCT reduction leads to a long and narrow westerly (easterly) anomaly over the Asian continent north (south) of 40°N at 200-hPa, encompassing the South and East Asia, whilst the ABS reduction yields the opposite changes. The change of Asian jet at 200-hPa caused by SCT reduction induces a

broad high-level divergence over South and East Asia, which motivates the moisture convergence in the whole layer and low-level upward motion (Fig. 12e and 12f). Therefore, an increased monsoon precipitation is found over South (Fig. 4c and 4g) and East Asia (Fig. 5c and 5g). In addition, located in the central part of the South Asia, monsoon trough is the portion of the intertropical convergence zone that extends into a monsoon circulation. The monsoon-trough-controlled area is usually the region with heavy precipitation due to its resulting cyclonic vorticity at lower level (Mishra et al., 2012). According to the

configuration of the enhanced divergence at the upper level (Fig. 11d) and increased vertical velocity (Fig. 12e), the monsoon trough is enhanced, thus creating strong dynamic conditions for precipitation over northern South Asia.

The ABS reduction-induced easterly anomalies to the north and westerly anomalies to the south of 40°N in contrast generates the high-level convergence anomalies over northern South Asia and East Asia (Fig. 11e). The resulting moisture divergence in the whole layer and low-level downward motions over northern South Asia and most of East Asia hinder the regional

monsoon precipitation (Figs. 11g and 11h). Following the impacts of the SCT reduction, emission reduction in total aerosols also contributes to the northward shift of the Asian jet, but the westerly anomaly north of 40°N and the accompanying upper-tropospheric divergence are weaker than that of reducing the SCT alone.

Fig. 13 compares the SAH changes induced by the short-term emission reductions in different aerosol types. The SAH gets stronger and moves northward when only the SCT is reduced, which could be attributed to the land warming over Asia. The

land cooling induced by the ABS reduction leads to a weaker SAH, but the SAH change is incomparable to that with the SCT-reduction. Owing to the dominant effects of the SCT reduction, reduction in total aerosols enhances the SAH intensity over Asia. Using reanalysis and observation data, Wei et al. (2015; 2017; 2021) conducted a series of studies and revealed that the northwestward (southeastward) shift of the SAH is closely related to more (less) Indian summer monsoon precipitation, more (less) north and south China precipitation and less (more) Yangtze River valley precipitation. The SAH intensity is also

positively related with monsoon precipitation over India, as reported in their studies. Along with the reduction of total or SCT aerosols and the resulting enhanced and northward moving SAH, the spatial pattern of summer precipitation anomalies over South and East Asia (Fig. 4 and 5) are consistent with their results. The easterly anomalies to the south of the enhanced SAH center (with the climatological value of more than 16800 gpm at 100-hPa) cause a stronger vertical wind shear over the northern

South Asia and southern East Asia due to the total or SCT reductions. The change in vertical wind shear has important impacts on the convective development by altering the dynamic instability (Wang et al., 2019). As a result, the intensity of the local convective systems is increased due to the strengthened dynamic instability over South and East Asia.

Besides, it is found that the linear addition can capture the feature of geopotential height (Fig. 9), upper-tropospheric jet (Fig. 11), moisture divergence field (Fig. 12) and SAH (Fig. 13) adjustments over Asia induced by total aerosols reduction.

## 4 Conclusions and discussions

The urgent need to mitigate global climate and environmental issues is likely to force drastic GHGs and aerosols emission reductions at the global scale through policy controls and technological innovations in the coming decades. The aerosol-forced temperature and precipitation anomalies dominate the global response when both emissions of carbon dioxide and aerosol were reduced (Fyfe et al., 2021). Hence, the potential signals and the effect mechanisms of short-term air pollution mitigation on the SASM and EASM in terms of the temporal extent and intensity are investigated based on UKESM1 simulations forced by reducing global aerosol emissions by a substantial fraction. The respective responses of different properties of the SASM and EASM to emission reductions in SCT and ABS aerosols are presented. The monsoon sensitivities to simultaneous and non-simultaneous emission reductions of the SCT and ABS aerosols are discussed.

There exists a large degree of similarity in the SASM and EASM responses to the aerosol emission reductions either in temporal or spatial scale. The SCT reduction-induced tropospheric warming and ABS reduction-induced tropospheric cooling over SASM- and EASM-controlled regions happens throughout the year. The warming induced by the SCT reduction over South and East Asia during pre- and post-monsoon seasons favors early transition of land-sea thermal contrast and SLP difference in spring and late transition in autumn, thus extending the monsoon by advancing its onset and delaying the withdrawal. The change in pressure gradient force induced by SCT aerosol reduction leads to an increase in westerlies to the north of the upper-tropospheric jet center, leading to the northward displacement of the high-level easterly and westerly jet. The northward displacement of the high-level jet causes the anomalous moisture convergence and upward motion at the lower level over north India and east China, eventually enhancing the precipitation over South and East Asia during monsoon season. The stronger SAH due to the land warming induced by the reduction of SCT also facilitates the local convective development over northern South Asia and southern East Asia. However, ABS reduction acts in the opposite sense in Asian climate responses, which delays the transition of land-sea contrast in spring and advancing the transition in autumn, forces the Asian jet to move southward, and weakens the SAH intensity.

Overall, reduction in SCT makes the rainy season over South and East Asia longer and stronger, while reduction in ABS makes the rainy season shorter and weaker. The aerosol-reduction-induced monsoon intensity changes over South and East Asia are dominated by the impacts of reducing SCT, while the onset and withdrawal dates adjustments of the SASM and EASM are controlled by the combined impacts of reducing SCT and ABS aerosols.

The spatial features of the linear summation of the individual effect from reducing SCT or ABS alone is similar to the effect of reducing both aerosol types simultaneously. However, differences in details between the linear summation and the results of reducing total aerosols indicates some non-linearity in the system as a whole. Various complex nonlinear interactions in the atmosphere (the mixing states of the SCT and ABS aerosols, the nonlinear changes in cloud fields induced by activated aerosols and other feedback from atmospheric thermal and dynamic processes) could contribute to the deviation. The difference of the

monsoon precipitation and circulation anomalies related to the atmospheric adjustments between the results from linear addition and the simulation with total aerosol reduction is more pronounced over South Asia compared to that over East Asia, indicating that the climate adjustments over South Asia show higher degrees of non-linear additivity. However, the non-linearity hardly affects the general pattern of the Asian monsoon and monsoon-related large-scale environmental adjustments caused by short-term aerosol emission reductions. Considering the unpredictable technological progress and policies, the

emission reduction pathways of scattering and absorbing aerosol components are possibly non-synchronous. The opposite adjustments of Asian rainy season forced by scattering and absorbing aerosol emission control and the performance of their linear summation need to be considered during the climate and environment policy-making process.

    In this study, we have attempted to investigate the signal of possible responses of different monsoon systems to a hypothesized air pollution mitigation by reducing aerosol emissions globally by a substantial fraction (75%). Although the emission

perturbations we apply here are hypothetical and are likely to be larger than those that will actually be implemented, our results illustrate the possible maximum extent of monsoon adjustments that emission reduction of different aerosol types might be expected to induce. Previous research has estimated that the annual global emissions of BC, $SO_2$ and organic carbon in 2100 are projected to decline by 64-91%, 56-74% and 8-30%, respectively, compared to 2015 in SSP1-1.9, SSP2-4.5 and SSP3-7.0 scenarios (Lund et al., 2019). Hence, the variation range of plausible responses of the SASM and EASM in terms of duration

and intensity may be similar or smaller than the simulated responses presented here.

    Additionally, bias may exist in the results of monsoon response due to the model performance in reproducing the monsoonal characteristics. General circulation models are often noted to have biases in the seasonal means of monsoon features (such as the precipitation). Jain et al. (2019) showed that the CMIP5 models show a prominent dry bias over northern and central SA in summer. The RMSD values of the simulated summer monsoon precipitation over SA land range from 3.50 to 8.54 mm day[-1]

among the CMIP5 models with respect to the observations in their evaluations. However, CMIP5 models tend to overestimate the precipitation in most regions of China, and the RMSD of the annual mean precipitation for the multi-model means is 3.98 mm day[-1] relative to the GPCC observations (Chen and Frauenfeld, 2014). The higher daily mean precipitation amount may lead to higher RMSD values in China if the evaluation is only conducted in boreal summer. UKESM1 is a CMIP6 era model that was developed from the CMIP5 era HadGEM2-ES model. The precipitation biases of CMIP6 and CMIP5 models align

closely at the spatiotemporal scale, though CMIP6 models show an improvement in reducing the precipitation bias in the Yangtze River valley, part of North China, Western Ghats and North-East foothills of Himalayas (Gusain et al., 2020; Xin et al., 2020). Here, we summarize the RMSD between the UKESM1 results and ERA5 reanalysis/observation in Fig. S2. Consistent with the CMIP5's bias on precipitation shown in previous research, UKESM1 yields an overall overestimation over

EA but underestimation over SA. The RMSD values reach maximum during monsoon season over EA (1.37 - 1.76 mm day$^{-1}$)
and SA (4.05 - 4.13 mm day$^{-1}$). The simulated bias of UKESM1 for monsoon precipitation over SA is at the lower end of the RMSE range from CMIP5 models, and the overestimation over EA is also lower than the multi-model means. Tian et al. (2021) pointed out that the UKESM1 is one of the CMIP6 models that exhibits better reproduction of historical precipitation over China. In addition, the signal of possible monsoon responses shown in this study are estimated by subtracting the aerosol-emission-perturbed runs from control runs by assuming that the systematic error in both the control and the aerosol-emission-perturbed simulations remains the same, and this assumption is inherent in most climate change studies.

Moreover, the positive climate change signal in Asian monsoon precipitation as well as the enhanced circulation in the future due to total aerosol reduction shown in this study is qualitatively consistent with the findings of previous research either focusing on the short-term impacts of COVID-19 lockdown (Kripalani et al., 2021) or long-term impacts of future emission scenarios (Zhao et al., 2018; Wilcox et al., 2020). The possible Asian monsoon adjustments regulated by reduction in SCT/ABS component further examined in this study are also the direct opposite of the SASM (Krishnamohan et al., 2021; Sherman et al., 2021) or EASM (Jiang et al., 2013; Xie et al., 2020) changes forced by the industrial SCT/ABS emission increase. Besides, the definitions used in this study for the EASM (W2016 and G1983) and SASM (W2009) have been validated the ability to show the monsoon onset for the historical period in previous research (Fang et al., 2020; Khandare et al., 2022). The N2016 index has also been verified to show consistent seasonal evolution with other dynamic and thermodynamic variables of the SASM (Noska and Misra, 2016). Based on the Community Atmosphere Model version5.1, Wang, D. et al (2016) showed an EASM onset delay and withdrawal advance caused by the SCT, and vice versa for the ABS. Kripalani et al. (2021) found that the summer monsoon withdrawal over India was delayed in 2020, which could be associated with the reduced aerosol during COVID-19 lockdown. All these findings support the signals of short-term air pollution mitigation on the SASM and EASM adjustments in terms of the temporal extent shown in this study.

The uncertainties due to the internal climate variability in the model should also not be ignored although we have tried to narrow the uncertainties by conducting an ensemble of ten stochastically perturbed simulations. These limitations given above indicate that the variational range of the simulated responses of the SASM and EASM duration and intensity presented here should be interpreted with caution. More climate projections for the response scales of the SASM and EASM duration and intensity under different aerosol emission pathways are needed in the future, but our simulations do suggest that a more comprehensive understanding of the impacts of aerosols on the monsoon systems can be achieved by separating aerosol absorbing and scattering components.

Numerical experiments are conducted by implementing the same level of reductions in the emissions of SCT, ABS and total aerosol. The summer climate adjustments over Asia are controlled by the impacts of the SCT, although are counterbalanced by the opposite changes induced by the ABS to some extent. The relative weaker response of the direct radiative forcing to the evolution of BC emissions has been reported by Ocko et al. (2014) based on a coupled atmosphere-ocean National Oceanic and Atmospheric Administration Geophysical Fluid Dynamics Laboratory global climate model. Liu et al. (2009) also pointed out that the forcing from BC aerosols over Asia is relatively weak and limited. Our findings indicate that the SCT reduction

will bring greater changes in Asian monsoon if the same emission control is implemented on the SCT and ABS aerosols. It is plausible that aerosol-cloud-interactions force some of this response; a reduction in SCT by 75% (mainly from sulfate and organic aerosols) will lead to a greater reduction in cloud-condensation nuclei than a reduction in absorbing aerosols by 75% (mainly from BC). While our work implicitly contains these impacts, future work should examine the attribution between aerosol-radiation interactions and aerosol-cloud-interactions.

This work focusses on the impacts of global reductions of SCT and ABS aerosols to examine the potential dynamical feedbacks and impacts on monsoon characteristics. A further area of research that is not pursued here is the role of local reductions of aerosol emissions (i.e. in the areas of investigation) versus reductions in aerosol concentrations outside of the areas of investigation. While this is outside the scope of this paper, further work is suggested in this area to better understand the role of changes in local versus remote aerosol emissions.

In addition, the slow climate feedback associated with the SST under the reduction of aerosol emissions cannot be represented in the simulation results due to the short simulation time conducted in this study. Wang et al. (2019) examined the fast and slow responses of Asian monsoon to anthropogenic aerosol forcings and found similar responses of the SASM and EASM to increasing total aerosols, manifesting as a robust drying trend and weakened monsoon circulation. However, they pointed out that the SASM adjustments are dominated by the SST change, while the EASM adjustments are largely due to the fast direct atmospheric response to aerosol radiative forcing. Therefore, we may infer that the long-term adjustments of the SASM and EASM caused by the emission reductions in total aerosols will still be similar, presenting by the enhanced monsoon circulation and increased precipitation, although the dominant mechanisms in regulating the SASM and EASM may be different.

*Code and data availability*. The observational precipitation data from the Climate Prediction Center (CPC) unified gauge-based daily observations (Chen et al. 2008) and the Global Precipitation Climatology Project (GPCP) rain gauge-satellite combined precipitation dataset (Huffman and Bolvin, 2013) used in this study can be obtained from https://psl.noaa.gov/data/gridded/data.cpc.globalprecip.html (last access: 2nd Jan. 2022) and https://www.ncei.noaa.gov/data/global-precipitation-climatology-project-gpcp-daily/access/ (last access: 15th Feb. 2022), respectively. The precipitation and wind fields from the ECMWF's (European Center for Medium-Range Weather Forecast) Fifth-generation Reanalysis (ERA5; Hersbach et al., 2020) are available at https://cds.climate.copernicus.eu/cdsapp#!/search?type=dataset&text=ERA5 (last access: 15th Feb. 2022). The CMIP6 historical simulations of the UKESM1 model are available at https://esgf-node.llnl.gov/search/cmip6/ (last access: 11th Feb. 2022). The model outputs and codes can be accessed by contacting Chenwei Fang via fangcw515@163.com.

*Author contributions*. The majority of this work was completed when CF was visiting the University of Exeter in the UK under the scholarship from China Scholarship Council. JMH proposed the idea. He also supervised this work and revised the

manuscript together with JL. BTJ performed the UKESM1 simulations. JL, YC and BZ provided useful suggestions for the study. All authors made contributions to the writing of this study.

*Competing interests.* The authors declare that they have no conflict of interest.

*Acknowledgments.* We are grateful to the University of Exeter for providing the academic platform for this study and Met Office for doing the numerical calculations in this work.

*Financial support.* This work is financially supported by the National Natural Science Foundation of China (42021004, 42192512). JMH and JL would like to acknowledge support from the NERC funded SASSO standard grant (NE/ S00212X/1).

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

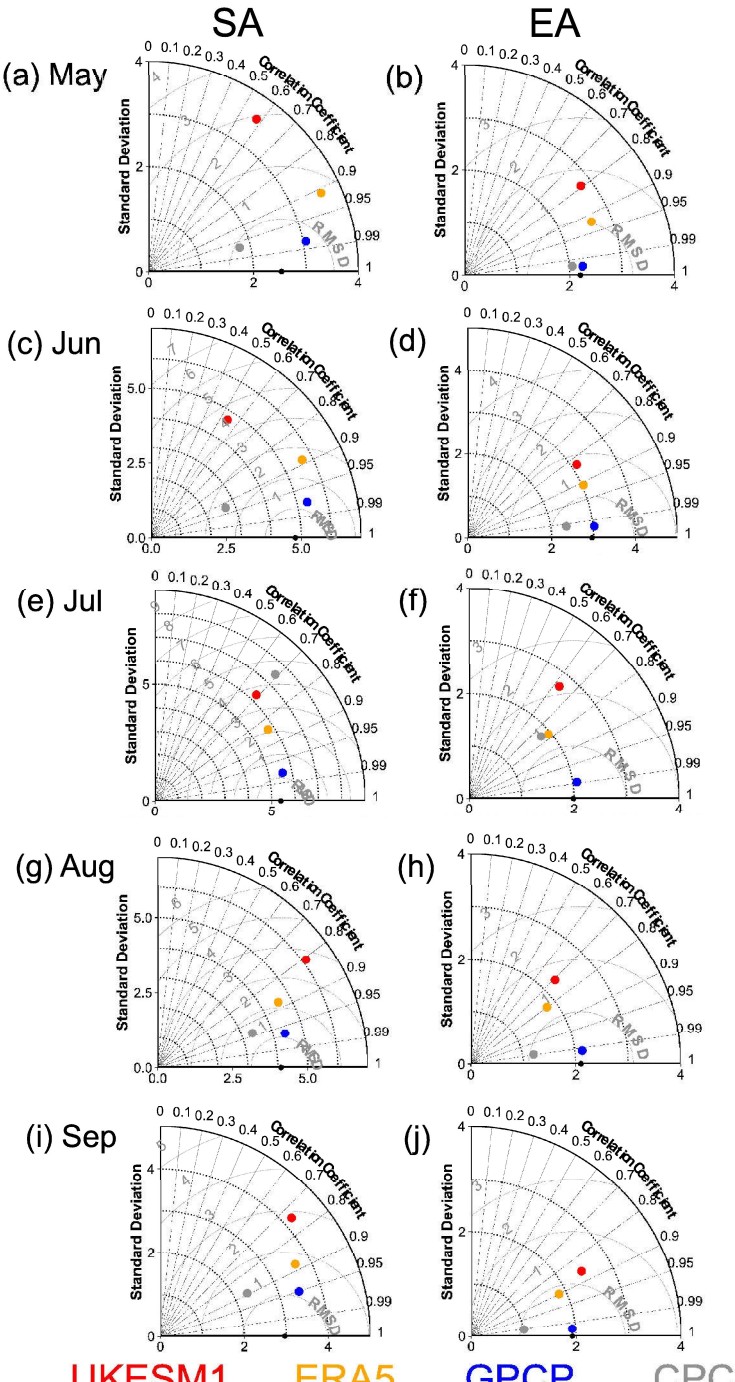

**Figure 1: Taylor diagrams for the simulated climatological monthly mean (1985-2014) summer precipitation (unit: mm day$^{-1}$) over South Asia (a, c, e, g and i) and East Asia (b, d, f, h and j) from the CMIP6-UKESM1 historical simulation (red dots), ERA5 reanalysis (yellow dots), GPCP dataset (blue dots) and CPC observations (gray dots). The merged observed mean values of the GPCP and CPC datasets are shown as black dots. The angular co-ordinate gives the correlation with the mean values of observations. The radial co-ordinate gives the standard deviations of different datasets. The dotted gray lines represent the root-mean-square deviation (RMSD).**


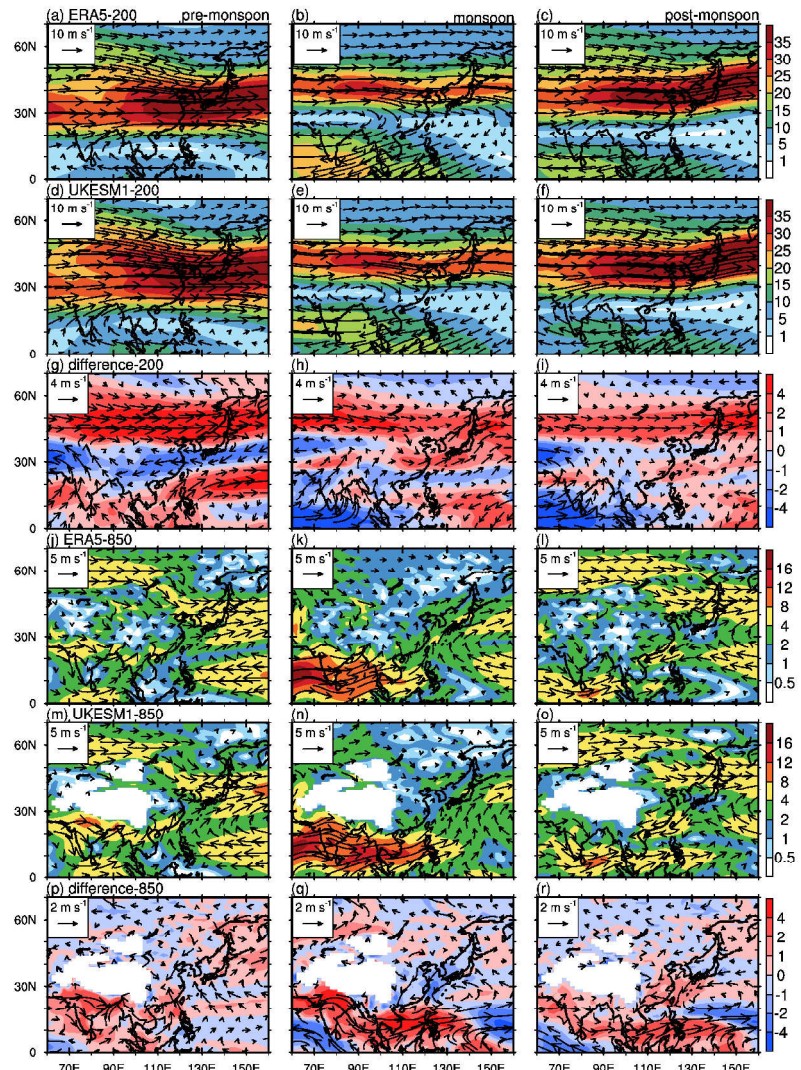

**Figure 2: Spatial distributions of the climatological mean (1985-2014) wind directions (vectors; unit: m s⁻¹) and wind speeds (shading; unit: m s⁻¹) at 200 hPa (a-i) and 850 hPa (j-r) from ERA5 reanalysis (a-c and j-l) and CMIP6-UKESM1 historical simulation (d-f and m-o) over Asia during pre-monsoon (April-May; a, d, g, j, m and p), monsoon (June-August; b, e, h, k, n and g) and post-monsoon (September-October; c, f, i, l, o and r) seasons. Panels (g-i) and (p-r) show the differences between the wind fields from the UKESM1 simulation and ERA5 reanalysis at 200 hPa and 850 hPa, respectively.**


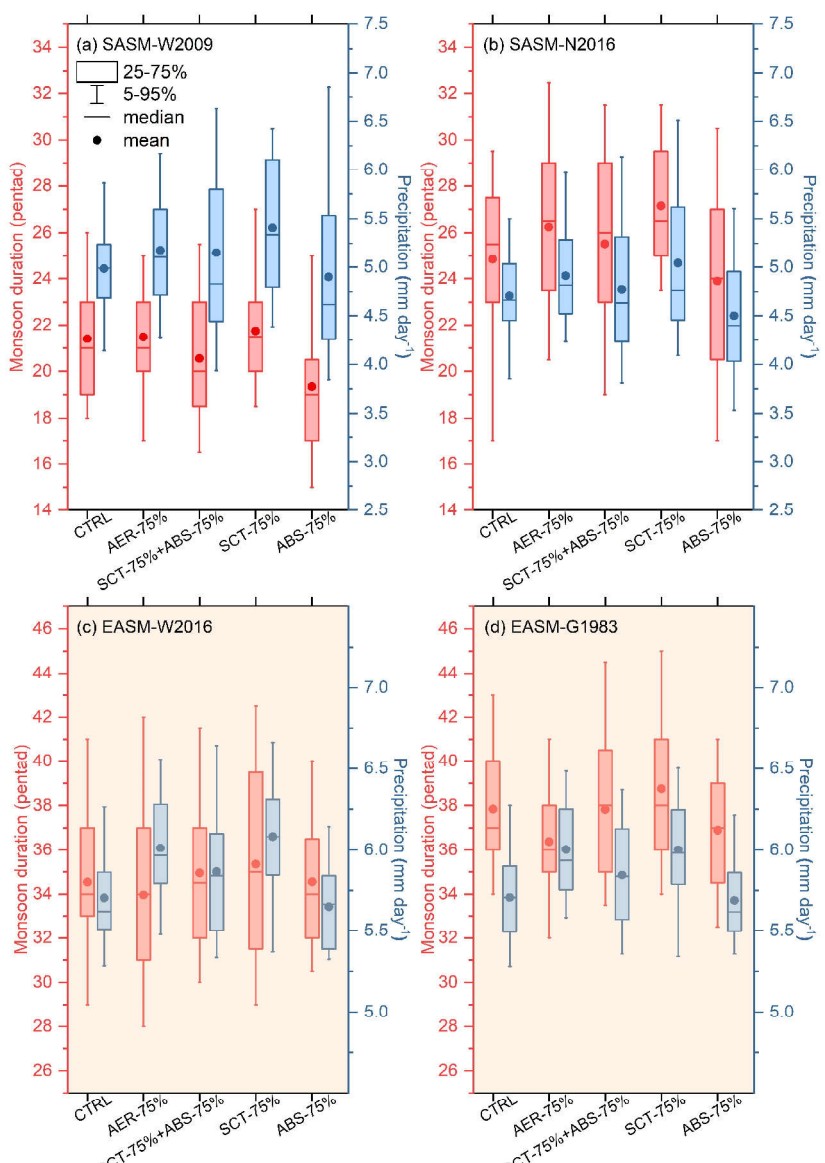

**Figure 3: Box diagrams of the monsoon duration (red; unit: pentad) and precipitation (blue; unit: mm day⁻¹) over South Asia (a and b) and East Asia (c and d) in different simulations. Dots and middle horizontal lines inside boxes indicate mean and median values, respectively, and lower and upper sides of boxes indicate 25 and 75% range, respectively, and top and bottom line represent 5% and 95%, respectively. The boxes labelled SCT-75+ABS-75% in each panel are the linear addition of the impacts of the reductions in the SCT and ABS. Panel (a) is derived based on the definition from Wang et al. (2009; hereafter referred to as W2009). Panel (b) is derived based on the definition from Noska and Misra (2016; hereafter referred to as N2016). Panel (c) is derived based on the definition from Wang, D. et al (2016; hereafter referred to as W2016). Panel (d) is derived based on the definition from Guo (1983; hereafter referred to as G1983).**

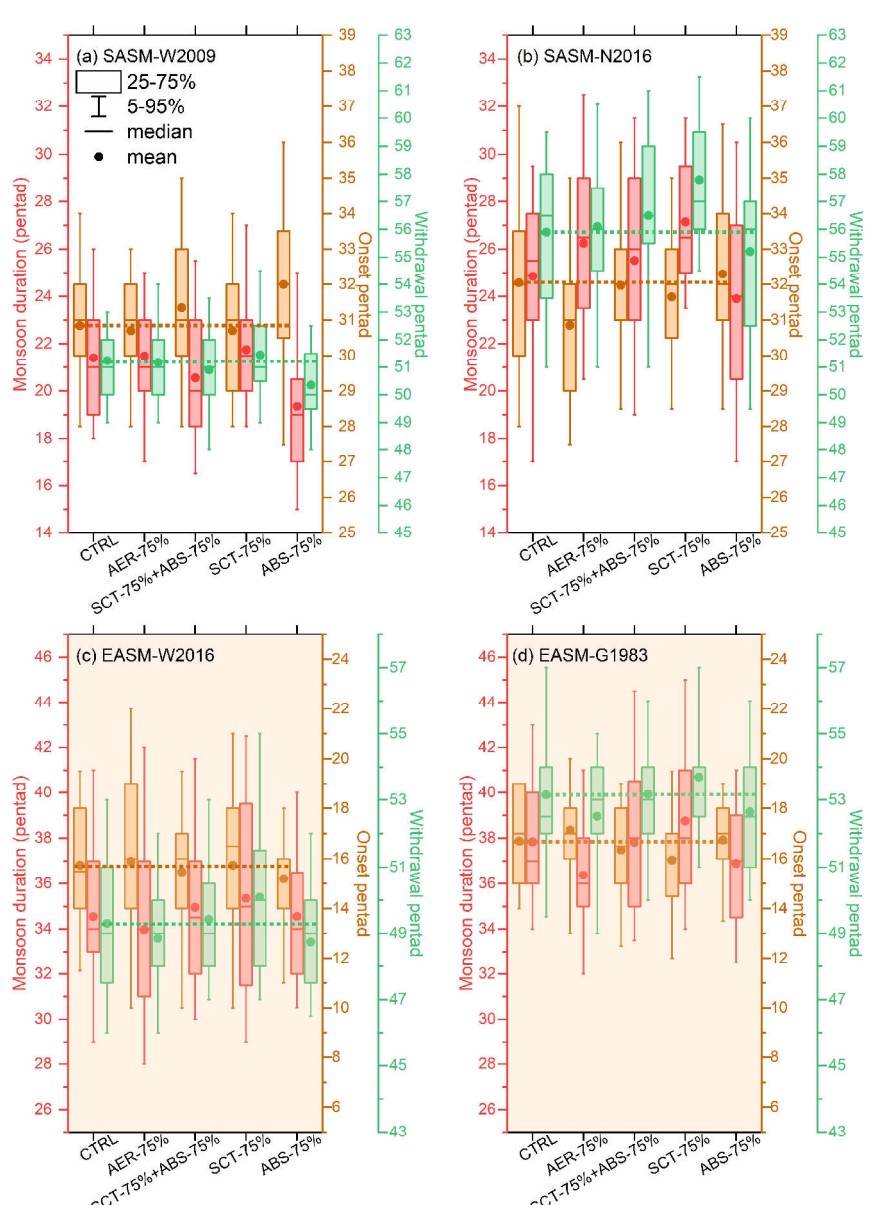


**Figure 4: Same as Figure 3, but for the monsoon onset dates (yellow; unit: pentad), withdrawal dates (green; unit: pentad) and duration (red; unit: pentad). The yellow and green dashed lines denote the mean values of monsoon onset and withdrawal in the CTRL simulation set, respectively.**


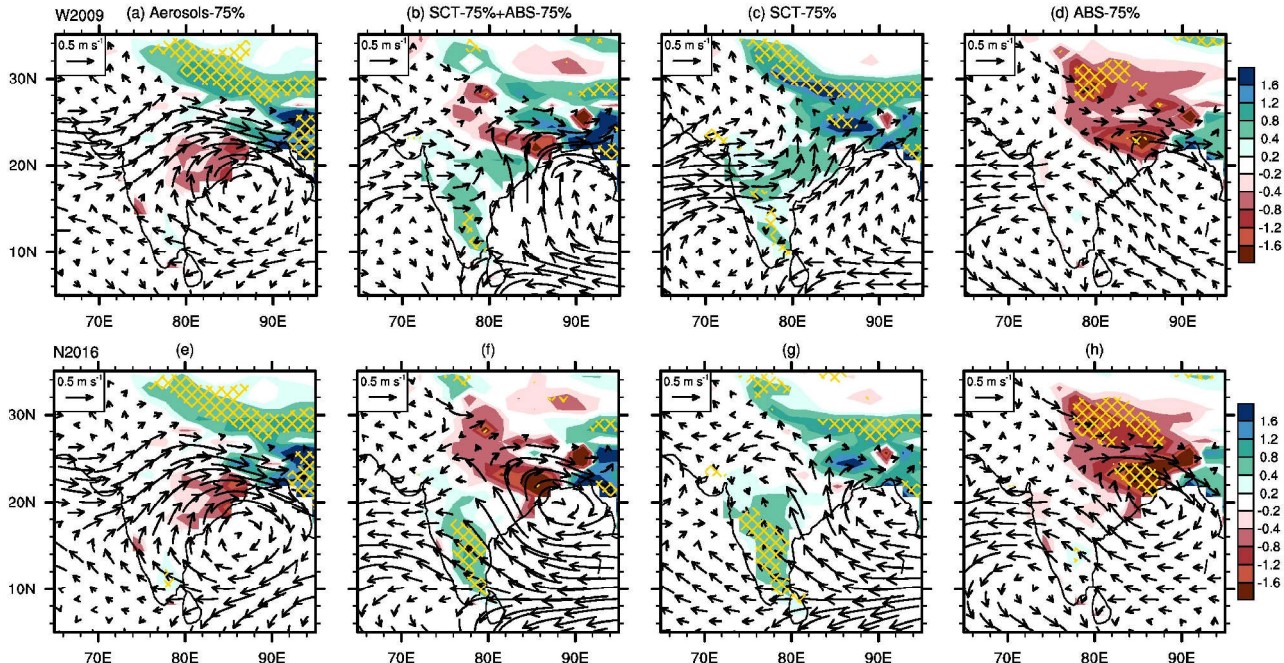

**Figure 5: Spatial distributions of the monsoon precipitation (shading; unit: mm day-1) and 850-hPa wind fields (vector; unit: m s⁻¹) responses to the reductions in total aerosols (a and e), scattering aerosols (SCT; c and g) and absorbing aerosols (ABS; d and h) over South Asia. Panels (b) and (f) are the linear addition of the impacts of the reductions in the SCT and ABS. Hatched regions denote where the precipitation change is statistically significant at the 95% confidence level according to a Wilcoxon rank sum test. Panels (a)-(d) are derived based on the definition from W2009. Panels (e)-(h) are derived based on the definition from N2016. The SASM adjustments forced by emission reductions in different aerosol types are the difference between the aerosol-emission-perturbed and control runs.**

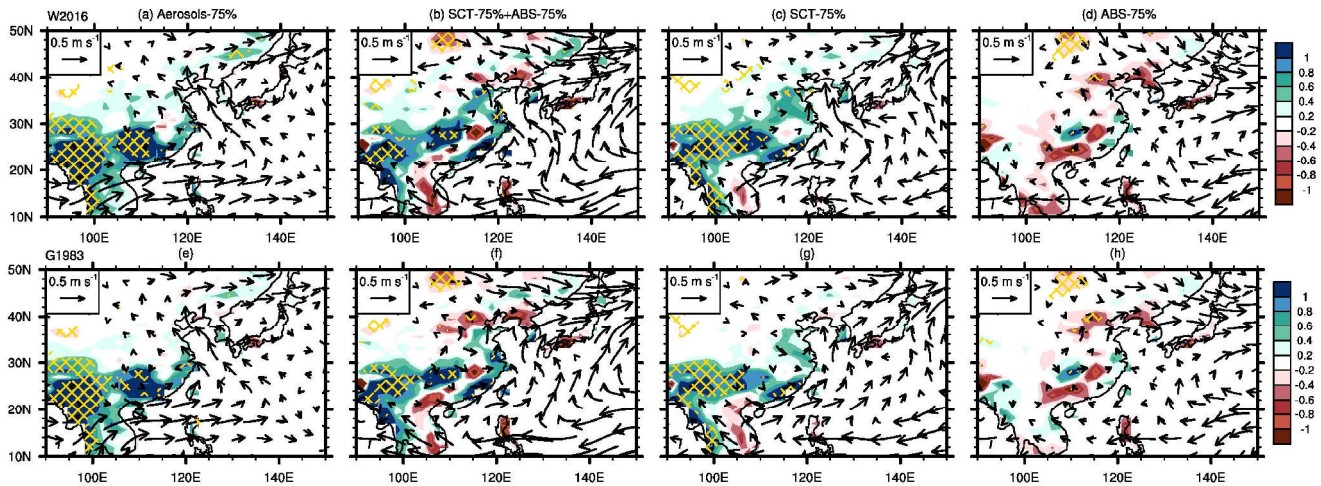

Figure 6: Same as Figure 5, but for East Asia. Panels (a)-(d) are derived based on the definition from W2016. Panels (e)-(h) are derived based on the definition from G1983.

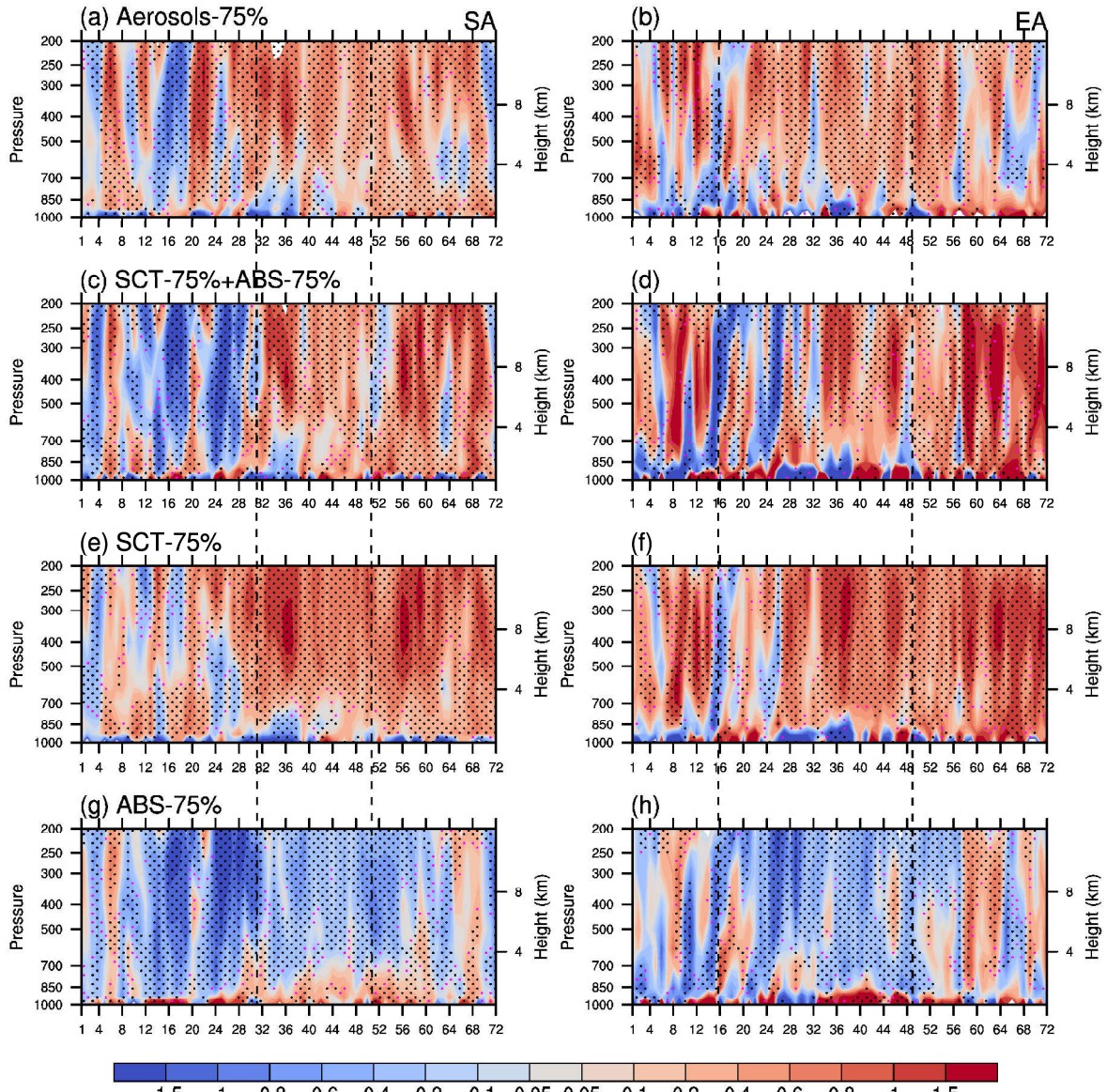

**Figure 7:** Time-altitude cross sections of the temperature (unit: K) responses to the reductions in total aerosols (a and b), SCT aerosols (e and f) and ABS aerosols (g and h) averaged over South Asia (a, c, e and g) and East Asia (b, d, f and h). The temperature responses are the difference between the aerosol-emission-perturbed and control runs. The x-axis denotes the time (unit: pentad). The division of South and East Asia used here follows Iturbide et al. (2020) and is also shown in Fig. S1. Panels (c) and (d) are the sum of the impacts of the reductions in the SCT and ABS. The first and second vertical dashed lines in Panels (a), (c), (e) and (g) denote the monsoon onset and withdrawal pentad over South Asia in the control experiment based on the definition from W2009. The first and second vertical dashed lines in Panels (b), (d), (f) and (h) denote the monsoon onset and withdrawal pentad over East Asia in the control experiment based on the definition from W2016. Black and pink dotted regions denote where the temperature change is statistically significant at the 95% and 90% confidence level, respectively, according to a t-test.

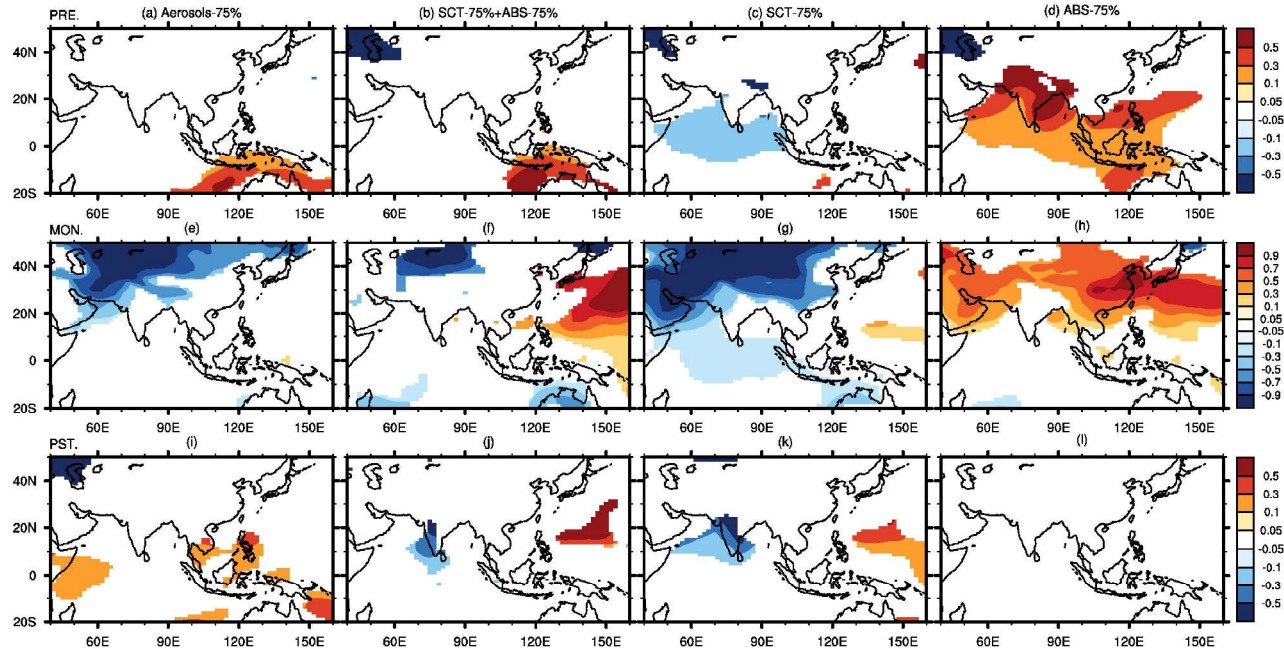

**Figure 8: Spatial distributions of the sea level pressure (unit: hPa) responses to the reductions in total aerosols (a, e and i), SCT aerosols (c, g and k) and ABS aerosols (d, h and l) over Asia during pre-monsoon (April-May; a-d), monsoon (June-August; e-h) and post-monsoon (September-October; i-l) seasons. The sea level pressure responses are the difference between the aerosol-emission-perturbed and control runs. Panels (b), (f) and (j) are the sum of the impacts of the reductions in the SCT and ABS. Only the sea level pressure changes with a confidence level of 95% or 90% according to the t-test are shown.**


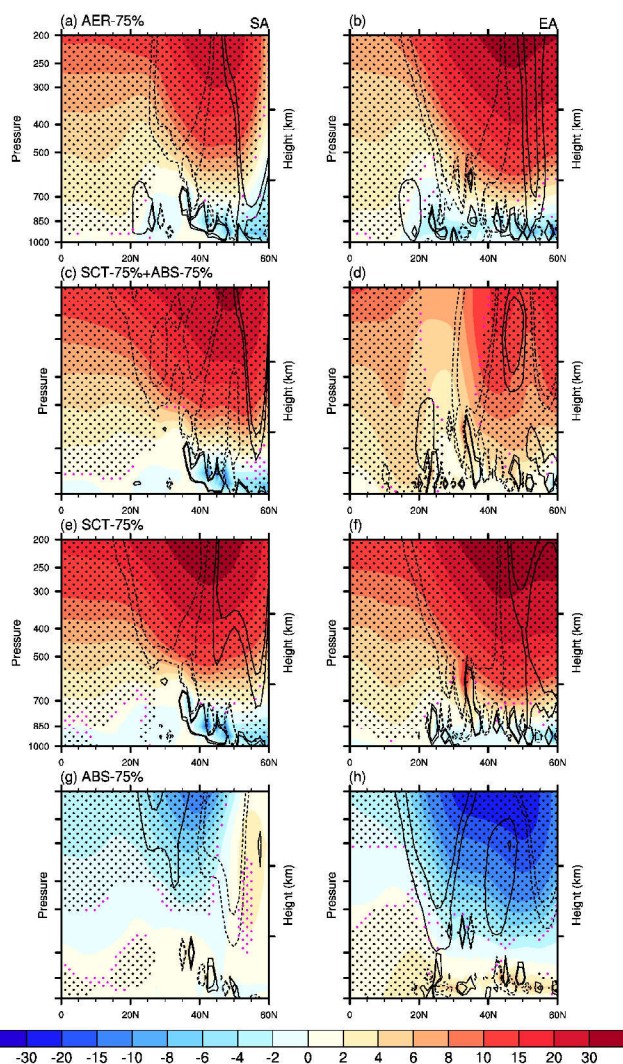


**Figure 9: Zonal-mean geopotential height (unit: gpm) responses to the reductions in total aerosols (a and b), SCT aerosols (e and f), and ABS aerosols (g and h) during monsoon season over South Asia (70-90°E; a, c, e and g) and East Asia (100-120°E; b, d, f, h). Monsoon season is analyzed and based on the definitions from W2009 over South Asia and W2016 over East Asia. The geopotential height responses are the difference between the aerosol-emission-perturbed and control runs. Panels (c) and (d) are the sum of the**
**impacts of the reductions in the SCT and ABS. Black lines denote the meridional gradient of GH response (unit: gpm m⁻¹; solid and dashed lines denote positive and negative values, respectively). Black and pink dotted regions denote where the geopotential height change is statistically significant at the 95% and 90% confidence level, respectively, according to a t-test.**

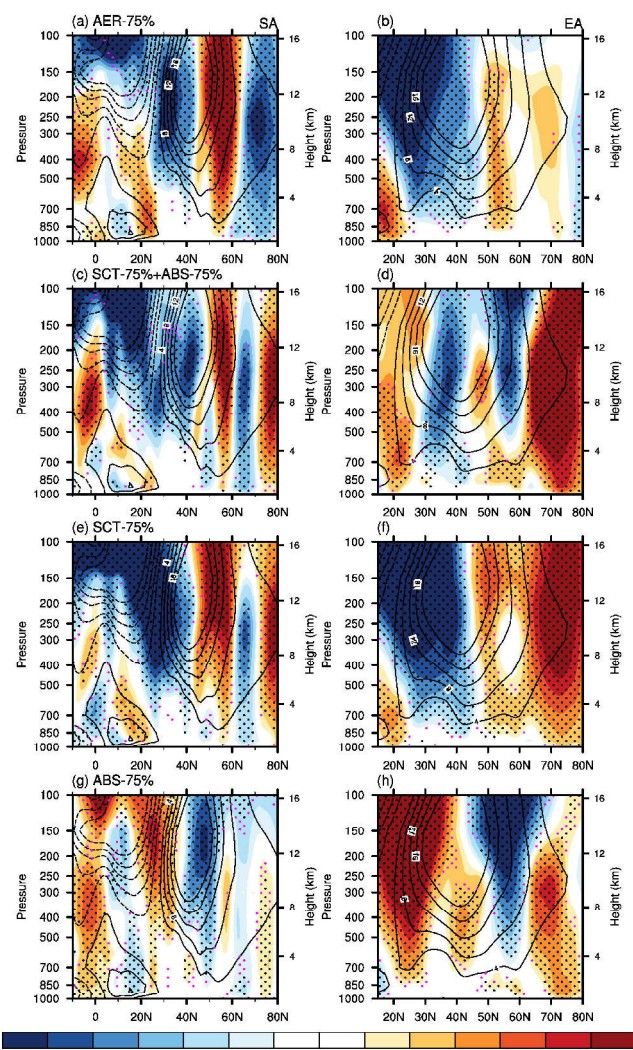

**Figure 10: Same as Figure 9, but for the zonal-mean zonal wind (shading; unit: m s⁻¹; red and blue denote westerly and easterly wind, respectively) responses during monsoon season. Monsoon season is analyzed and based on the definitions from W2009 over South Asia and W2016 over East Asia. Black lines represent the climatological zonal wind from control simulations (unit: m s⁻¹; solid and dash lines denote westerly and easterly wind, respectively).**

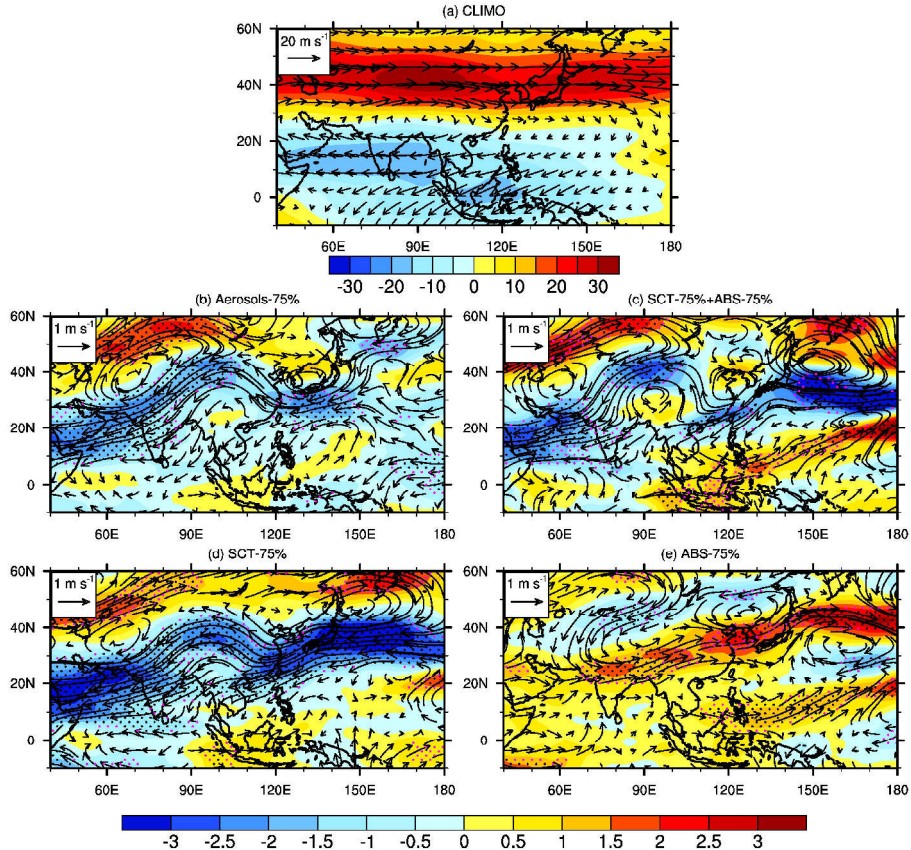

**Figure 11: Spatial distributions of the 200-hPa zonal wind (shading; unit: m s$^{-1}$) and wind fields (vectors; unit: m s$^{-1}$) responses to the reductions in total aerosols (b), SCT aerosols (d) and ABS aerosols (e) over Asia during monsoon season (June-August). Panel (a) is the climatological 200-hPa zonal wind (unit: m s$^{-1}$) and wind fields (unit: m s$^{-1}$) from control simulations. The 200-hPa wind fields responses are the difference between the aerosol-emission-perturbed and control runs. Panel (c) is the linear addition of the impacts of the reductions in the SCT and ABS. Black and pink dot-ted regions denote where the zonal wind change is statistically significant at the 95% and 90% confidence level, respectively, according to a t test.**

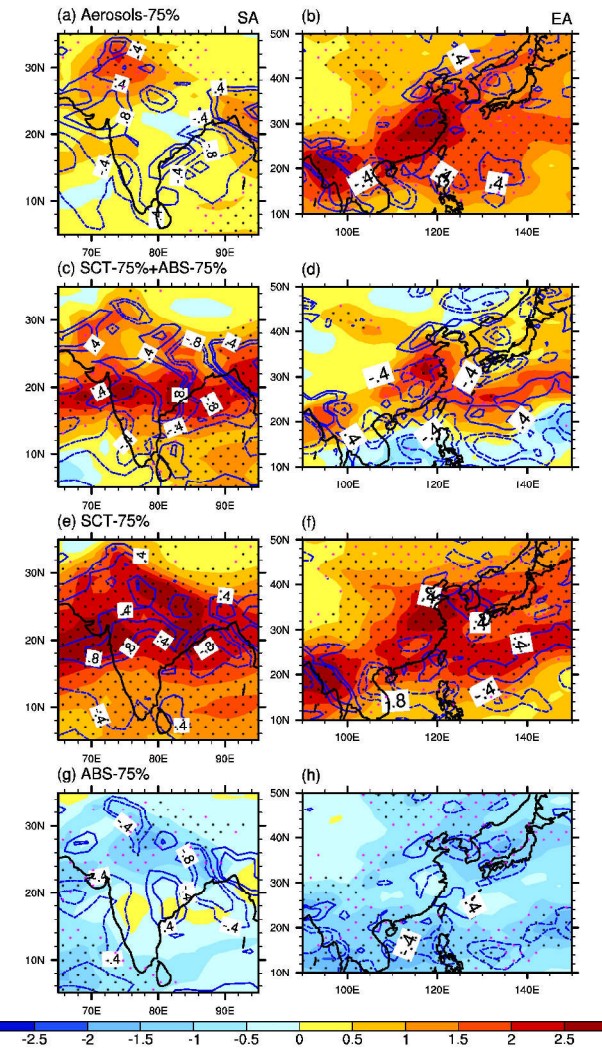


**Figure 12: Spatial distributions of the total column moisture flux (shading; unit: kg m$^{-2}$ s$^{-1}$; red denote moisture convergence and blue denote divergence) and 850-hPa vertical velocity (contours; unit: -100×Pa s$^{-1}$) responses to the reductions in total aerosols (a and b), SCT aerosols (e and f) and ABS aerosols (g and h) during monsoon season over South Asia (a, c, e and g) and East Asia (b, d, f and h). Monsoon season is analyzed and based on the definitions from W2009 over South Asia and W2016 over East Asia. The total column moisture flux and 850-hPa vertical velocity responses are the difference between the aerosol-emission-perturbed and control runs. Panels (c) and (d) is the linear addition of the impacts of the reductions in the SCT and ABS. Black and pink dotted regions denote where the total column moisture flux change is statistically significant at the 95% and 90% confidence level, respectively, according to a t-test.**

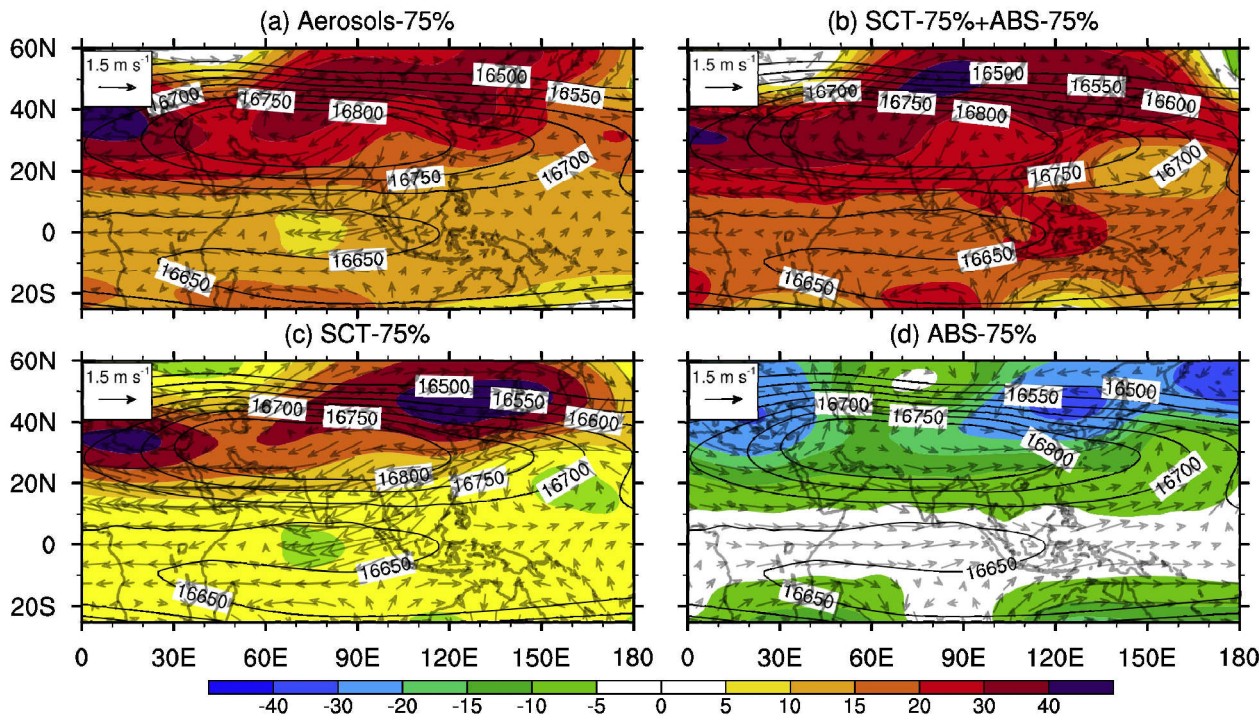

**Figure 13: Spatial distributions of the 100-hPa geopotential height (shading; unit: gpm) and wind fields (vectors; unit: m s$^{-1}$) responses to the reductions in total aerosols (a), SCT aerosols (c) and ABS aerosols (d) over Asia during monsoon season (June-August). The 100-hPa geopotential height and wind fields responses are the difference between the aerosol-emission-perturbed and control runs. Panel (b) is the linear addition of the impacts of the reductions in the SCT and ABS. Black lines is the climatological geopotential height (unit: m) from control simulations, and the center value of South Asian High is more than 16800 gpm.**