# Peer review of "Impacts of reducing scattering and absorbing aerosols on the temporal extent and intensity of South and East Asian summer monsoon"

_EGUsphere, 2023_

## Referee Comment (RC2)

In this study, the authors examined the response of SASM and EASM to idealized reductions in anthropogenic emissions of carbonaceous and $SO_2$ under SSP3-7.0 scenario. The authors found that reducing both scattering and absorbing aerosols intensifies monsoon systems. The overall impact is dominated by scattering aerosols, while absorbing aerosols have the opposite effects. The reduction in scattering aerosols advances the monsoon onset but delays its withdrawal. The experiments are well designed, and results are clearly presented. I have a few comments for the authors to consider.

General Comments:
1. I would suggest the authors add more discussions, especially for section 3.1 and 3.2, to clarify a bit more. The authors tend to only describe the figures very briefly and don't give much explanation (dynamical mechanism) to the changes induced by reductions in scattering and absorbing aerosols. Most of the content for Fig. 3 is introduction of the method. Maybe the authors can put method/calculation related content to section 2 so that there is more room for detailed discussion. The interpretation is blended with method, which is easy to get lost. There is also little discussion related to Fig. 4 and 5.

2. It is interesting to see the linearity/non-linearity when combing reductions in both scattering and absorbing aerosols. However, I would give a second thought about discussing this mostly in the last result section (sect. 3.4). The authors can either blend this section with other sections and give an overall discussion in the conclusion section or at least say a few words about the linearity in other sections.

3. How would the model biases, such as in precipitation and monsoon onset/withdraw, affect the signals from perturbed simulations? For example, the overview paper of UKESM1 by Stellar et al. (2019) shows considerable low biases of precipitation in JJA over South Asia.

4. Have the authors looked at the impact of reducing local (with SA and EA) anthropogenic emissions versus the impact of reducing anthropogenic emissions outside of the two regions? How much of the changes in SASM and EASM are induced by the reduction of local anthropogenic emissions?

Specific comments:
Lines 116-117, what do you mean by "includes the physical core climate model of"? Could you rephrase it?

Lines 143-144, just would like to check that anthropogenic emissions of $SO_2$, OM, and BC are reduced globally, not just for South and East Asia, right?

Line 156, what do you mean by "different random perturbations in the stochastic physics"? Perturbing temperature of the initial state? Could you be more specific?

Lines 178-179, I would suggest the authors mention Fig. S1 here or even before to explain the definition of the two domains.

Line 215, missing a space between $C_m$ and of.

Lines, 272-274, the conclusion seems to be too general, and may not be the situation for both EASM and SASM. For example, how about the increase of SLP over China in Fig. 8c?

Lines, 279-282, I would argue that Fig 8a seems to be more close to Fig 8c, indicating SCT dominating. Fig. 8i does not like Fig. 8j-l, indicating strong non-linearity?

Lines 295-300, should it be "atmospheric warming associated with the SCT reduction" according to Figure 7? Should it be "The geopotential height increases in the uppermost troposphere …" instead of "The pressure .."? I would argue that it is not so obvious in Figure 7 that the geopotential height changes strengthens the poleward pressure gradient force in the north flank of this area.

Line 345, should be "may not be a linear summation of"?

Lines 384-393, I would suggest the authors state the impacts for SCT and ABS in two separate sentences instead of putting antonyms in parentheses. It is easy to get lost when reading long sentences.

Figure 2, I would suggest changing the title for middle column to be MON to be consistent with PRE and PST. Or maybe it is better to just pre-monsoon, monsoon, and post-monsoon.

Figure 3, could you change scale for left y-axis of panel b from day to pentad? It may be clearer to compare panel a and b. Is it possible to add values showing linearly combined SCT-75% and ABS-75%?

Figure 6, similar as Figure 3, could you change scale of panel b from day to pentad for better comparison and consistency?

---

## Author Comment (AC1)

Dear Referee,

Thanks for giving us an opportunity to revise our manuscript (ID: egusphere-2023-407). We appreciate your positive and constructive comments. We have studied these comments carefully and make revisions on the manuscript. We believe that the manuscript has benefited substantially from this revision with much clearer presentation. These com- ments and the corresponding replies are listed below.

The reviewer's comments are highlighted by gray. The symbol "≫" quotes the original texts in the manuscript. Fol- lowed by the comments are our responses (normal texts) and current texts in the manuscript (**leaded by line number**

**in the manuscript with the tracked changes**). Some important revisions are colored by red.

With regards,

Chenwei Fang*, Jim M. Haywood, Ju Liang, Ben T. Johnson, Ying Chen and Bin Zhu*

**Replies to Referee#1**

**1.** Line 33 - the sentence starting with "Our findings suggest that…" is hard to read and unclear. Please rephrase it.

≫Line 33: Our findings suggest that emission controls that target e.g. emissions of black carbon that warm the climate would have a different response to those that target overall aerosol emissions.

We have corrected this sentence, as shown below:

**Lines 33-34**

The opposing adjustments of Asian rainy season forced by the ABS and SCT emission reduction suggest that emission controls that target e.g. emissions of black carbon that warm the climate would have a different response to those that target overall aerosol emissions.

**2.** Line 140: The authors mentioned that SSP3-7.0 represents a high baseline climate with strong pollution, which is obviously true. However, since the simulations included in this study stops in 2024, I guess the emission levels be- tween SSP3-7.0 and the other SSP scenarios are largely the same, which may be worth noting.
* * *
≫Line 140: Hence, the control simulations based on SSP3-7.0 scenario represent a high baseline from which to assess
the maximum climate response to strong pollution mitigation.

Thank you for this valuable suggestion. We have corrected this sentence.

**Lines 142-144**

However, the simulations in this study stop in 2024, when aerosol emissions in SSP3-7.0 are still close to those in oth-
er SSP scenarios according to the estimation of Lund et al. (2019). Hence, the control simulations based on SSP3-7.0
scenario give a reasonable baseline prediction of the period assuming typical levels of emissions persist.

**3.** Line 196: In addition to the positive difference between the model and ERA5 north of 40N, the negative difference
at around 30N (jet core) should also be emphasized. Effectively, the model-simulated jet is wider but less intensive
compared to observation, especially during the pre-monsoon season, if I read Fig2g correctly.

≫Line 196: However, it should be noted that the simulated upper-level westerly jet northward of 40°N from pre- to
post-monsoon seasons are stronger compared to ERA-5 reanalysis (Fig. 2g-i).

Thank you for this valuable suggestion. We agree that the descriptions about the difference in upper-level jet between
the model results and ERA5 are unclear. The relevant description has been added.

**Lines 234-237**

However, it should be noted that the simulated upper-level westerly jet shows a positive difference northward of 40°N
and a negative difference around 30°N from pre- to post-monsoon seasons compared to ERA-5 reanalysis (Fig. 2g-i),
indicating a slightly wider but less intensive westerly jet in the UKESM1 simulation, especially for the pre-monsoon
season.

**4.** Line 199 - "Southerly wind prevailing over East Asia is slightly underestimated": this is unclear to me: which region
is mentioned here? Please clarify.

≫Line 199: The lower-level southwest monsoon flow over South Asia is also overestimated in the model, while the monsoon southerly wind prevailing over East Asia is slightly underestimated (Fig. 2q).

We have corrected this sentence, as shown below:

**Lines 237-239**

The lower-level southwest monsoon flow over South Asia is also overestimated in the model, while the monsoon southerly wind prevailing over East Asia between 20 and 40°N is slightly underestimated (Fig. 2q).

**5.** Paragraph starting at line 218: The discussion on the box plots needs more clarification from my perspective. The authors argued that W2009 and N2016 share similar statistical results, but I feel less confident about this. For example, the SCT-driven extension of the SASM duration is less obvious based on W2009 (Fig3a). I would recommend adding more quantitative descriptions, and including more comparisons between the two methods as well as explaining the possible reasons.

≫ Paragraph starting at line 218: The SASM duration and precipitation in Fig. 3(a) and (b) show similar changes, although they are based on different definitions. Compared to the SASM in the control case, reduction in SCT extends the temporal extent of the SASM duration and enhances the monsoon precipitation, while reduction in ABS shortens the SASM and reduces the monsoon precipitation. With the combined effects induced by SCT and ABS aerosols, reduction in total aerosols has negligible impacts on the temporal extent of SASM and enhances the monsoon precipitation although the enhancement is weaker than pure SCT reduction.

Thank you for this valuable suggestion. We agree that using different definitions of monsoon onset/withdrawal dates may result in the variations in the monsoon duration response range although the SASM duration and precipitation adjustments in W2009 and N2016 are qualitatively consistent. According to your suggestions, we (1) changed the scale for left y-axis of Fig. 3(b; N2016) from day to pentad for better comparison and consistency between Fig. 3a (W2009) and 3b (N2016); (2) added more quantitative descriptions and make a comparison between the Fig. 3(a) and (b) in Lines 271-286; (3) showed the possible causes for the difference in the SASM duration response between W2009 and N2016 in Lines 271-286.

**(1) Lines 988-996 (Figure 3)**

[Figure]

**Figure 3: Box diagrams of the monsoon duration (red; unit: pentad) and precipitation (blue; unit: mm day$^{-1}$) over South Asia (a and b) and East Asia (c and d) in different simulations. Dots and middle horizontal lines inside boxes indicate mean and median values, respectively, and lower and upper sides of boxes indicate 25 and 75% range, respectively, and top and bottom line represent 5% and 95%, respectively. The boxes labelled SCT-75+ABS-75% in each panel are the linear addition of the impacts of the reductions in the SCT and ABS. Panel (a) is derived based on the definition from Wang et al. (2009; hereafter referred to as W2009). Panel (b) is derived based on the definition from Noska and Misra (2016; hereafter referred to as N2016). Panel (c) is derived based on the definition from Wang, D. et al (2016; hereafter referred to as W2016). Panel (d) is derived based on the definition from Guo (1983; hereafter referred to as G1983).**

Note that using different definitions of monsoon onset/withdrawal dates may result in the variations in the monsoon duration response range although the SASM duration and precipitation adjustments in W2009 and N2016 are qualitatively consistent. The SASM durations in N2016 from different simulation sets are basically 4-5 pentads longer than those in W2009 (Fig. 3a and b). The SCT-driven extension of the SASM duration based on N2016 (2 pentads) is also longer than that in W2009 (0.4 pentads). The difference in the SASM duration adjustments between W2009 and N2016 can be attributed to the distinct selection of monsoon feature to characterize the monsoon subseasonal variations. Syroka et al (2004) pointed out that the withdrawal of the SASM defined by the precipitation is much later than that defined by the monsoon circulation due to the late decrease in precipitation in southern India. The precipitation continues to increase in southern Indian after September associated with the winter monsoon (Bhanu Kumar et al., 2004), while the SASM-related circulation characteristics becomes unclear in the meantime. Therefore, the SASM onset dates based on N2016 is roughly the same with those based on W2019, but the withdrawal date is about 5 pentads later, resulting in the longer monsoon duration (Fig. 4a and b). Moreover, there exists an additional enhancement of monsoon precipitation over SA in the "SCT" set, which further leads to the later SASM withdrawal and longer SASM duration in N2016 (Fig. 4b). Besides, the precipitation during early autumn is sensitive to the location and synoptic/sub-synoptic systems (tropical cyclones, depressions, easterly waves, north-south trough activity and coastal convergence, etc; Bhanu Kumar et al., 2004), which possibly contributes to the larger variation range in the monsoon withdrawal date in N2016.

[Figure]

**Figure 4: Same as Figure 3, but for the monsoon onset dates (yellow; unit: pentad), withdrawal dates (green; unit: pentad) and duration (red; unit: pentad). The yellow and green dashed lines denote the mean values of monsoon onset and withdrawal in the CTRL simulation set, respectively.**

**6.** Fig4 and other contour figures: How are the precipitation and wind response calculated? Are they the difference between control runs and aerosol-cut runs? Please clarify this in either the caption or method section.

Sorry, we didn't make it clear. The precipitation and wind response are the difference between aerosol-cut runs and control runs. The relevant description has been added in the method section and caption of Fig. 5-13.

For example:

**Lines 156-157 (Methods)**

The Asian monsoon adjustments forced by pollution mitigation are diagnosed as the difference between the aerosol-emission-perturbed and control runs.

**Lines 1011-1012 (caption of Fig. 5)**

……The SASM adjustments forced by emission reductions in different aerosol types are the difference between the aerosol-emission-perturbed and control runs.

**7.** Fig4c & g: the wind responses over the Indian Ocean (10N-20N) are quite different between W2009 and N2016, which also significantly affect the patterns in Fig4b&f. Can you explain the possible reasons and guess which method is potentially better representing the general structure of SASM? I suggest mentioning this issue in the relative paragraph and adding some discussions.

Thank you for your remind. We have added a paragraph (Lines 309-324) mentioning the wind response difference between W2009 and N2016. In this paragraph, (1) Lines 309-321 includes the relevant description and the possible causes for the wind response difference between W2009 and N2016; (2) Lines 321-324 gives the advantages and disadvantages of W2009 and N2016 in defining the SASM onset and withdrawal. **Note that in the revised manuscript, the serial number of Fig. 4 is changed to Fig. 5.**

**Lines 309-324**

Fig. 5 and 6 show the spatial patterns of the opposing changes in the precipitation and the 850-hPa circulation of the SASM and EASM induced by the SCT and ABS reductions. The monsoon precipitation changes over SA are consistent in W2009 and N2016, showing significantly increased (decreased) precipitation due to SCT (ABS) reduction. The low-level SASM circulation is also enhanced (weakened) over the Indian peninsular with the reduced SCT (ABS) based on W2009, while the wind response is different in N2016. The wind field adjustment in N2016 is characterized by a weak southwesterly anomaly over the north-central part of the Arabian Sea (north of 20°N) but an easterly anomaly over the south India and south Arabian Sea (10-20°N). The enhancement of easterly flow over SA could be associated with the relatively late monsoon withdrawal dates (58th pentad; Table S1) based on N2016 in the "SCT" set. The continuously increasing precipitation related to winter monsoon in the southern part of SA after September (Syroka et. al, 2004) and the SCT-reduction-induced increased precipitation in SA (Fig. 4g) jointly lead to the delay of the SASM withdrawal date to October based on N2016. At this time, the low-level circulation over south SA and south Arabian Sea is dominated by the prevailing easterly (October-December; Sengupta and Nigam, 2019) although the local precipitation remains elevated, and is associated with the summer monsoon precipitation according to the N2016. Hence, the onset is better defined than the withdrawal based on the precipitation definition adopted in N2016, especial-ly over southern SA. The W2009 definition is more widely applicable over SA, and the summer monsoon precipitation increase is more logically coherent with the circulation enhancement based on this definition.

[Figure]

Figure 5: Spatial distributions of the monsoon precipitation (shading; unit: mm day⁻¹) and 850-hPa wind fields (vector; unit: m s⁻¹) responses to the reductions in total aerosols (a and e), scattering aerosols (SCT; c and g) and absorbing aerosols (ABS; d and h) over South Asia. Panels (b) and (f) are the linear addition of the impacts of the reductions in the SCT and ABS. Hatched regions denote where the precipitation change is statistically significant at the 95% confidence level according to a Wilcoxon rank sum test. Panels (a)-(d) are derived based on the definition from W2009. Panels (e)-(h) are derived based on the definition from N2016. The SASM adjustments forced by emission reductions in different aerosol types are the difference between the aerosol-emission-perturbed and control runs.

**8.** Line 240: according to Fig5d&h, The decreased precipitation induced by ABS seems to be insignificant. Please double check and clarify the descriptions.

≫Line 240: …… Reduction in ABS mainly induces a decrease of precipitation in different subregions over East Asia.

Thank you for this valuable suggestion. The relevant description has been modified. Note that in the revised manu-
script, the serial number of Fig. 5 is changed to Fig. 6.

**Lines 328-331**

Reduction in ABS mainly induces a decrease of precipitation in different subregions over East Asia, but the decrease is
significant only in the regions with large changes. There are also some regions with increased anomalous precipitation,
but most of the precipitation increase was not statistically significant in this study ($p < 0.05$).

[Figure]

Figure 6: Same as Figure 5, but for East Asia. Panels (a)-(d) are derived based on the definition from W2016. Panels (e)-(h) are
derived based on the definition from G1983.

**9.** Line 242: The SCT seems to only dominate the precipitation over the north-eastern regions in Fig4, while the de-
crease in precipitation at around 20N seems to be related to ABS, are nonlinear effect between ABS and SCT. Please
double check and clarify.

≫ Line 242: In general, the impacts of the SCT reductions dominate both the SASM and EASM adjustments related to
the monsoon precipitation and circulation changes induced by short-term total aerosols mitigation.

Thank you for this valuable suggestion. The relevant description has been modified in Lines 336-340. Note that in the revised manuscript, the serial number of Fig. 4 is changed to Fig. 5 (see 7[th] Reply).

**Lines 336-340**

In general, the impacts of the SCT reductions dominate both the SASM and EASM adjustments related to the monsoon precipitation increase and circulation enhancement induced by short-term total aerosols mitigation. It should be noted that the SCT reduction only dominate the precipitation increase over the north-eastern SA (north of 22°N) induced by the total aerosol reduction. The precipitation decrease over central SA (south of 22°N) is contributed by the impacts of ABS reduction and non-linear effects between the SCT and ABS, but the decrease is insignificant in both W2009 and N2016 (Fig. 5).

**10.** Paragraph starting at line 244: Similar to my previous concerns about the box plot. W2009 seems to show very small differences between control runs and aerosol-cut runs (e.g., the SASM duration in the SCT run). Do the differences mentioned in this paragraph pass the significance test? Please clarify.

≫ Paragraph starting at line 244: To determine the SASM and EASM duration changes, the variations in monsoon onset and withdrawal dates are further examined (Fig. 6). The mean values and the 25th-75th percentile ranges of the monsoon onset date, withdrawal date and duration over South and East Asia are also summarized in Table S1. Reduction in SCT advances the SASM onset but delays the SASM withdrawal, thus extending the SASM duration to a certain extent (0.4 pentads in W2009 and 11.4 days in N2016)……

Thank you for your remind. We have added a paragraph to discuss the difference between W2009 and N2016 you mentioned in this comment and in the 5[th] comment. **Note that in the revised manuscript, the serial number of Fig. 6 showing the monsoon onset and withdrawal responses to aerosol reductions is changed to Fig. 4.**

We agree that W2009 seems to show a very small difference in the SASM duration between the SCT-reduction runs and control runs compared to that in N2016 (Fig. 4a and b). According to your suggestions in this comment and in the 5[th] comment, we (1) change the scale for left y-axis of Fig. 4b (N2016) from day to pentad for better comparison and consistency between Fig. 4a (W2009) and 4b (N2016); (2) add more quantitative descriptions and make a comparison between the Fig. 4(a) and (b) in Lines 271-286; (3) show the possible causes for the longer SASM duration and later

SASM withdrawal responses in N2016 compared to those in W2009 in Lines 271-286. **Note that the added texts in**

**Lines 269-282 has also been shown in the 5th Reply.**

**(1) Lines 999-1002 (Fig. 4)**

[Figure]

**Figure 4: Same as Figure 3, but for the monsoon onset dates (yellow; unit: pentad), withdrawal dates (green; unit: pentad) and**
**duration (red; unit: pentad). The yellow and green dashed lines denote the mean values of monsoon onset and withdrawal in the**
**CTRL simulation set, respectively.**

**(2)-(3) Lines 271-286 (The differences between the Fig. 4(a) and (b) and the possible causes)**

Note that using different definitions of monsoon onset/withdrawal dates may result in the variations in the monsoon duration response range although the SASM duration and precipitation adjustments in W2009 and N2016 are qualitatively consistent. The SASM durations in N2016 from different simulation sets are basically 4-5 pentads longer than those in W2009 (Fig. 3a and b). The SCT-driven extension of the SASM duration based on N2016 (2 pentads) is also longer than that in W2009 (0.4 pentads). The difference in the SASM duration adjustments between W2009 and N2016 can be attributed to the distinct selection of monsoon feature to characterize the monsoon subseasonal variations. Syroka et al (2004) pointed out that the withdrawal of the SASM defined by the precipitation is much later than that defined by the monsoon circulation due to the late decrease in precipitation in southern India. The precipitation continues to increase in southern Indian after September associated with the winter monsoon (Bhanu Kumar et al., 2004), while the SASM-related circulation characteristics becomes unclear in the meantime. Therefore, the SASM onset dates based on N2016 is roughly the same with those based on W2019, but the withdrawal date is about 5 pentads later, resulting in the longer monsoon duration (Fig. 4a and b). Moreover, there exists an additional enhancement of monsoon precipitation over SA in the "SCT" set, which further leads to the later SASM withdrawal and longer SASM duration in N2016 (Fig. 4b). Besides, the precipitation during early autumn is sensitive to the location and synoptic/sub-synoptic systems (tropical cyclones, depressions, easterly waves, north-south trough activity and coastal convergence, etc; Bhanu Kumar et al., 2004), which possibly contributes to the larger variation range in the monsoon withdrawal date in N2016.

**11.** Line 273 - "lowers SLP anomaly over Asian continent compared with that over Indian and western Pacific oceans": maybe worth noting the opposite changes over the Indian Ocean and western Pacific in Fig8g.

≫Line 273: The SCT reduction induced land warming yields a lower SLP anomaly over Asia continent compared with that over Indian and western Pacific oceans, which is favourable for the early/late transition of land-sea pressure difference in pre/post-monsoon season and a stronger SASM and EASM circulation in monsoon season.

≫Fig. 8:

[Figure]

**Figure 8: Spatial distributions of the sea level pressure (unit: hPa) responses to the reductions in total aerosols (a, e and i), SCT**
**aerosols (c, g and k) and ABS aerosols (d, h and l) over Asia during pre-monsoon (April-May; a-d), monsoon (June-August; e-h)**
**and post-monsoon (September-October; i-l) seasons. Panels (b), (f) and (j) are the sum of the impacts of the reductions in the SCT**
**and ABS. Black and pink dotted regions denote where the sea level pressure change is statistically significant at the 95% and 90%**
**confidence level, respectively, according to a t-test.**

Thank you for this valuable suggestion. We have modified the Fig. 8 in order to show the changes in sea level pressure
(SLP) caused by aerosol emission reductions more intuitively. Only the SLP changes with a confidence level of 95%
or 90% according to the t-test are shown in the new Fig. 8 (Lines 1032-1038). We also added a new figure (Fig. S3 in
Supplement) to quantitatively examine the anomalous land-sea SLP difference between the Asian continent and its
surrounding oceans and seas. The descriptions and discussions about the opposite changes over the Indian Ocean and
western Pacific in the new Fig. 8(g) are added in Lines 396-405.

**Lines 1032-1038 (new Figure 8)**

[Figure]

**Figure 8: Spatial distributions of the sea level pressure (unit: hPa) responses to the reductions in total aerosols (a, e and i), SCT aerosols (c, g and k) and ABS aerosols (d, h and l) over Asia during pre-monsoon (April-May; a-d), monsoon (June-August; e-h) and post-monsoon (September-October; i-l) seasons. The sea level pressure responses are the difference between the aerosol-emission-perturbed and control runs. Panels (b), (f) and (j) are the sum of the impacts of the reductions in the SCT and ABS. Only the sea level pressure changes with a confidence level of 95% or 90% according to the t-test are shown.**

**Lines 48-56 (Fig. S3; Supplement with tracked changes)**

[Figure]

**Figure S3. Time series of the anomalous land-sea sea level pressure (SLP) difference (unit: hPa) between the Asian continent part**
**(including South Asia, East Asia, Tibet Plateau and East-Central Asia) adjacent to the ocean and its surrounding oceans and seas**
**(including Northwest Pacific, tropical Indian Ocean, Bay of Bengal and Arabian Sea) to the reductions in total aerosols (b; gray**
**line), SCT aerosols (a; red line) and ABS aerosols (a; blue line). The x-axis denotes the time (unit: pentad). The land-sea SLP dif-**
**ference responses are the difference between the aerosol-emission-perturbed and control runs. Purple line in Panel (b) represent**
**the sum of the impacts of the reductions in the SCT and ABS. The shading area denote the standard deviation of the land-sea SLP**
**difference anomaly. The sub-panel attached to Panel (a) gives the climatological land-sea SLP difference (unit: hPa) from control**
**simulations. The region division used in this study refers to the sixth IPCC assessment report and is shown in Fig. S1.**

**Lines 396-405**

The SCT reduction induces a negative land-sea SLP difference anomaly throughout the year (Fig. S3 and Fig. 8c, g and k), which is favourable for the advance in the land-sea SLP difference transition from positive to negative in spring and the delay in the transition from negative to positive in autumn. The negative anomalous land-sea SLP difference also leads to bigger land-sea SLP contrast and a stronger SASM and EASM circulation in the monsoon season. Note that the SLP changes in part of the oceanic areas adjacent to the SA region are consistent with the continental SLP

changes, albeit with a smaller range of decrease (Fig. 8c, g and k). This could potentially be attributed to the reduced ACT that transported from Asian continent. However, the SLP decrease over these oceanic areas exerts negligible influence on the overall SCT-reduction-induced anomalous negative land-sea SLP difference between the Asian continent and adjacent oceans (Fig. S3).

**12.** Line 277: The reduced land-sea pressure contrast is not shown in Fig8h, as both land and sea show increases in SLP.

≫Line 277: The reduced land-sea pressure contrast during monsoon season also weakens the Asian monsoon intensities.

Thank you for this valuable suggestion. Considering the increased SLP both over land and sea, we have added a new figure (Fig. S3 in Supplement; see 11[th] Reply) to quantitatively examine the anomalous land-sea SLP difference between the Asian continent and its surrounding oceans and seas. The ABS reduction induced a positive land-sea SLP difference anomaly during monsoon season (Fig. S3a), although the SLP increases over Asian land and part of its surrounding seas (Fig. 8h; see 11[th] Reply). Hence, the land-sea SLP contrast is reduced during monsoon season and weakens the Asian monsoon intensities because of the positive land-sea SLP difference anomaly.

**13.** Line 282: How do you get the conclusion that Fig8a & i show patterns with combined effects of SCT and ABS? Do you calculate the map correlation or any other regression methods? It seems to me that the pre and post-monsoon patterns are more complicated. For example, Fig8a is very similar to Fig8c but shows an insignificant pattern over the Indian Ocean. I would suggest a more careful statement here; otherwise, more analyses are necessary.

≫Line 282: In other seasons except summer, the land-sea SLP adjustments over Asia is controlled by the combined effects of SCT- and ABS-reductions (Fig. 8a and 8i).

Sorry, we didn't make it clear. We have added more analysis and made a more careful statement. Besides the new added Fig. S3 (Supplement; see 11[th] Reply; quantitatively examine the anomalous land-sea SLP difference induced by reductions in total, SCT and ABS aerosols reductions)  and modified Fig. 8 (see 11[th] Reply; only show the SLP changes with a confidence level of 95% or 90% according to the t-test to have a clearer presenting of SLP adjustments), we have added discussions about the Fig. 8 (a and i) to clarify the dominant effects in regulating the SLP responses over
Asian continent and its surrounding oceans and seas in Lines 411-421.

**Lines 411-421**

In addition, the anomalous land-sea SLP difference between the Asian continent and the topical Indian and Northwest
Pacific Oceans caused by the short-term total aerosols mitigation during monsoon season is dominated by the SCT
aerosols and enhances the monsoon circulation over South and East Asia (Fig. 8e). There is also a negative land-sea
SLP difference anomaly due to the total aerosols mitigation in pre- and post-monsoon seasons (Fig. S3b), which is
governed by the impacts of  SCT-reduction. However, the spatial pattern of SLP adjustments during pre-monsoon sea-
son induced by total aerosol reduction shows a SLP increase over the seas of Southeast Asia, and both the impacts of
SCT- and ABS-reduction (Fig. 8 c and d) contribute to the SLP increase over this region. Besides, the ABS-reduction
has no significant impacts on the SLP adjustments during post-monsoon season (Fig. 8i). But the regions with signifi-
cant SLP changes caused by total aerosol reduction are also inconsistent with those caused by the SCT reduction (Fig.
8i and k), indicating the strong non-linearity of atmospheric system.

**14.** Fig8: Are these values over land surface pressure instead of sea level pressure? Also, the color bar for panels e-h
should be extended since it is hard to see more detailed patterns in panels e and g.

The sea level pressure ($SLP$) shown in Fig. 8 refers to the concept of the "corrected pressure", in which the surface or
station pressure ($P$) is corrected to sea level by estimating the weight of an imaginary column of air that extends from
surface or station to sea level: $SLP = P + h\rho g$, where $h$ is height of the land surface or site above sea level, $\rho$ is the air
density and $g$ is the acceleration of gravity. In this way, the pressure in different areas can be compared without the
impacts of terrain. Besides, the color bar for panels e-h is extended in new Fig. 8 (see 11[th] Reply). Thank you for this
valuable suggestion.

---

## Author Comment (AC2)

Dear Referee,

Thanks for giving us an opportunity to revise our manuscript (ID: egusphere-2023-407). We appreciate your positive and constructive comments. We have studied these comments carefully and make revisions on the manuscript. We believe that the manuscript has benefited substantially from this revision with much clearer presentation. These com- ments and the corresponding replies are listed below.

The reviewer's comments are highlighted by gray. The symbol "≫" quotes the original texts in the manuscript. Fol- lowed by the comments are our responses (normal texts) and current texts in the manuscript (**leaded by line number**

**in the manuscript with the tracked changes**). Some important revisions are colored by red.

With regards,

Chenwei Fang*, Jim M. Haywood, Ju Liang, Ben T. Johnson, Ying Chen and Bin Zhu*

**Replies to Referee#2**

**General Comments:**

**1.** I would suggest the authors add more discussions, especially for section 3.1 and 3.2, to clarify a bit more. The au- thors tend to only describe the figures very briefly and don't give much explanation (dynamical mechanism) to the changes induced by reductions in scattering and absorbing aerosols. Most of the content for Fig. 3 is introduction of the method. Maybe the authors can put method/calculation related content to section 2 so that there is more room for detailed discussion. The interpretation is blended with method, which is easy to get lost. There is also little discussion related to Fig. 4 and 5.

Thank you for this valuable suggestion. We agree that the descriptions about the definition for monsoon onset and withdrawal are more like the research background of this study. There is also a lack of discussions related to Fig. 3-6.

According to your suggestions, we (1) moved the descriptions about the definition for monsoon onset and withdrawal to the Methods section and made it a sub-section (Lines 182-206); (2) added more quantitative descriptions and made a comparison between the SASM adjustments calculated based on the W2009 and N2016 in Lines 271-286, 309-324

(for Fig. 3-5); (3) added more quantitative descriptions and made a comparison between the EASM adjustments calcu- lated based on W2016 and G1983 in Lines 294-308, 333-335 (for Fig. 3, 4 and 6).

**(1) Lines 182-206 (Methods)**

[revised manuscript text omitted]

**(3) Lines 294-308 and Lines 333-335 (comparison between the EASM adjustments calculated based on the W2016 and G1983)**

Lines 294-308

The impacts of reducing SCT and ABS on the EASM in terms of timescale and intensity (here is characterized by precipitation amount) are similar to that on the SASM, except that the reduction in total aerosols slightly shortens the temporal extent of the EASM (more pronounced in G1983) and increases the summer precipitation over the EASM-controlled region (Fig. 3c and d). The monsoon duration is extended by about 1 pentad both in W2016 and G1983 due to the reduction in SCT, which is mainly from the monsoon withdrawal deferment (Fig. 4c and d). Reduction in ABS oppositely advances the withdrawal, leading to a shorter monsoon (1 pentad in G1983) in East Asia. Compared to the distinguishable EASM withdrawal adjustments, the SCT- or ABS-reduction induced EASM onset adjustments calculated by W2016 (based on meridional wind) and G1983 (based on land-sea pressure difference) are not obvious and consistent, indicating the complexity of EASM onset. He et al. (2008) also pointed out that the EASM exhibits a progressive and complicated establishment and a swift withdrawal. The EASM onset date is postponed but the withdrawal date is advanced due to the total aerosol reduction, hence the EASM temporal extent is shortened a little (0.5 pentads in W2016 and 1.4 pentads in G1983). Compared to the EASM adjustments in W2016, the EASM show longer duration (about 3 pentads) in G1983 due to the later withdrawal (about 4 pentads). Zhu et al. (2012) has clarified that the climatological transition date of the zonal land-sea contrast in autumn over EASM-controlled region is about 3 pentads later than that of the monsoon circulation, which largely explained the relatively late monsoon withdrawal dates in G1983.

Lines 333-335

……Overall, the EASM adjustments in terms of the temporal extent and intensity calculated based on G1983 are basically consistent with the results based on the W2016, in spite of the relatively late monsoon withdrawal dates in G1983, which adopts the land-sea pressure difference as the key monsoon characteristic.

**2.** It is interesting to see the linearity/non-linearity when combing reductions in both scattering and absorbing aerosols. However, I would give a second thought about discussing this mostly in the last result section (sect. 3.4). The authors can either blend this section with other sections and give an overall discussion in the conclusion section or at least say a few words about the linearity in other sections.

Thank you for this valuable suggestion. We have moved the discussions about the Asian monsoon responses in the linear addition and those in the simulation of reducing total aerosols to the end of each sub-section (Lines 356-380; Lines 423-428; Lines 487-488) and given an overall discussion in the conclusion section (Lines 548-563).

**Lines 356-380 (Section 3.2 Response of monsoon temporal extent and intensity)**

[revised manuscript text omitted]

**3.** How would the model biases, such as in precipitation and monsoon onset/withdraw, affect the signals from perturbed simulations? For example, the overview paper of UKESM1 by Stellar et al. (2019) shows considerable low biases of precipitation in JJA over South Asia.

Thank you for this valuable suggestion. We have added a figure (Fig. S2; Lines 27-33) in the Supplement presenting the UKESM1 bias in precipitation over Asia. We have also added the discussions about the impacts of model uncertainties in monsoon precipitation and onset/withdraw on our simulated results in the Conclusions and discussions Section (Lines 572-606).

**Lines 27-33 (Fig. S2; Supplement with tracked changes)**

[Figure]

Figure S2. Spatial distributions of the climatological mean (1985-2014) root-mean-square deviation (RMSD; unit: mm day$^{-1}$) between the simulations and ERA5 reanalysis (a-c) during pre-monsoon (April-May; a), monsoon (June-August; b) and post-monsoon (September-October; c) seasons. (d)-(f): Same as (a)-(c), but for the RMSD between the simulations and merged observations from Global Precipitation Climatology Project (GPCP) rain gauge-satellite combined precipitation dataset and Climate Prediction Center (CPC) unified gauge-based daily observations. The regional mean RMSD values over EA and SA are shown in blue and red text, respectively.

**Lines 572-606 (Conclusions and discussions)**

Additionally, bias may exist in the results of monsoon response due to the model performance in reproducing the monsoonal characteristics. General circulation models are often noted to have biases in the seasonal means of monsoon features (such as the precipitation). Jain et al. (2019) showed that the CMIP5 models show a prominent dry bias over northern and central SA in summer. The RMSD values of the simulated summer monsoon precipitation over SA land range from 3.50 to 8.54 mm day$^{-1}$ among the CMIP5 models with respect to the observations in their evaluations. However, CMIP5 models tend to overestimate the precipitation in most regions of China, and the RMSD of the annual mean precipitation for the multi-model means is 3.98 mm day$^{-1}$ relative to the GPCC observations (Chen and Frauenfeld, 2014). The higher daily mean precipitation amount may lead to higher RMSD values in China if the evaluation is only conducted in boreal summer. UKESM1 is a CMIP6 era model that was developed from the CMIP5 era HadGEM2-ES model. The precipitation biases of CMIP6 and CMIP5 models align closely at the spatiotemporal scale, though CMIP6 models show an improvement in reducing the precipitation bias in the Yangtze River valley, part of North China, Western Ghats and North-East foothills of Himalayas (Gusain et al., 2020; Xin et al., 2020). Here, we summarize the RMSD between the UKESM1 results and ERA5 reanalysis/observation in Fig. S2. Consistent with the

CMIP5's bias on precipitation shown in previous research, UKESM1 yields an overall overestimation over EA but underestimation over SA. The RMSD values reach maximum during monsoon season over EA (1.37 - 1.76 mm day$^{-1}$) and SA (4.05 - 4.13 mm day$^{-1}$). The simulated bias of UKESM1 for monsoon precipitation over SA is at the lower end of the RMSE range from CMIP5 models, and the overestimation over EA is also lower than the multi-model means. Tian et al. (2021) pointed out that the UKESM1 is one of the CMIP6 models that exhibits better reproduction of historical precipitation over China. In addition, the signal of possible monsoon responses shown in this study are estimated by subtracting the aerosol-emission-perturbed runs from control runs by assuming that the systematic error in both the control and the aerosol-emission-perturbed simulations remains the same, and this assumption is inherent in most climate change studies.

Moreover, the positive climate change signal in Asian monsoon precipitation as well as the enhanced circulation in the future due to total aerosol reduction shown in this study is qualitatively consistent with the findings of previous research either focusing on the short-term impacts of COVID-19 lockdown (Kripalani et al., 2021) or long-term impacts of future emission scenarios (Zhao et al., 2018; Wilcox et al., 2020). The possible Asian monsoon adjustments regulated by reduction in SCT/ABS component further examined in this study are also the direct opposite of the SASM (Krishnamohan et al., 2021; Sherman et al., 2021) or EASM (Jiang et al., 2013; Xie et al., 2020) changes forced by the industrial SCT/ABS emission increase. Besides, the definitions used in this study for the EASM (W2016 and G1983) and SASM (W2009) have been validated the ability to show the monsoon onset for the historical period in previous research (Fang et al., 2020; Khandare et al., 2022). The N2016 index has also been verified to show consistent seasonal evolution with other dynamic and thermodynamic variables of the SASM (Noska and Misra, 2016). Based on the Community Atmosphere Model version5.1, Wang, D. et al (2016) showed an EASM onset delay and withdrawal advance caused by the SCT, and vice versa for the ABS. Kripalani et al. (2021) found that the summer monsoon withdrawal over India was delayed in 2020, which could be associated with the reduced aerosol during COVID-19 lockdown. All these findings support the signals of short-term air pollution mitigation on the SASM and EASM adjustments in terms of the temporal extent shown in this study.

**4.** Have the authors looked at the impact of reducing local (with SA and EA) anthropogenic emissions versus the impact of reducing anthropogenic emissions outside of the two regions? How much of the changes in SASM and EASM are induced by the reduction of local anthropogenic emissions?

Thank you for this valuable suggestion. We agree that the issue of local versus global reductions in aerosol emissions and how the local versus aerosols transported into the region from other sources outside of the region of investigation is an interesting one. However, the focus of the paper would be considerably changed if we were to follow the suggestion of including additional simulations investigating this impact which would make the paper rather too long. We therefore suggest that the best way forward is to include a caveat in the discussion and conclusion (Lines 628-632) that highlights this issue as a potential area of future work.

**Lines 628-632 (Conclusions and discussions)**

This work focusses on the impacts of global reductions of SCT and ABS aerosols to examine the potential dynamical feedbacks and impacts on monsoon characteristics. A further area of research that is not pursued here is the role of local reductions of aerosol emissions (i.e. in the areas of investigation) versus reductions in aerosol concentrations outside of the areas of investigation. While this is outside the scope of this paper, further work is suggested in this area to better understand the role of changes in local versus remote aerosol emissions.

**Specific comments:**

**5.** Lines 116-117, what do you mean by "includes the physical core climate model of"? Could you rephrase it?

≫ Lines 116-117: The modelling system includes the physical core climate model of HadGEM3-GC3.1 (Hadley Centre Global Environment Model version 3; Kuhlbrodt et al., 2018; Williams et al., 2018) and the UKCA model (U.K. Chemistry and Aerosols model; Archibald et al., 2020; Mulcahy et al., 2018), along with terrestrial carbon and nitrogen cycles, dynamic vegetation and interactive ocean biogeochemistry.

We have rephrased this sentence, as shown below:

**Lines 117-121**

The modelling system is built on top of the core physical model HadGEM3-GC3.1 (Hadley Centre Global Environment Model version 3; Kuhlbrodt et al., 2018; Williams et al., 2018) and interactively coupled with the atmospheric components UKCA model (U.K. Chemistry and Aerosols model; Archibald et al., 2020; Mulcahy et al., 2018), terrestrial biogeochemistry and ocean biogeochemistry.

**6.** Lines 143-144, just would like to check that anthropogenic emissions of SO2, OM, and BC are reduced globally, not just for South and East Asia, right?

≫ Lines 143-144: In the "Total" set, all anthropogenic emissions of SO2, organic matter (OM) and BC were reduced by 75% relative to the SSP3-7.0 scenario.

Yes, aerosol emissions were perturbed globally. We have added the relevant descriptions in the Methods section.

**Lines 146-147**

To investigate the theoretical impacts that short-term pollution mitigation may have, three other sets of simulations were performed in which aerosol emissions were perturbed globally in different ways.

**7.** Line 156, what do you mean by "different random perturbations in the stochastic physics"? Perturbing temperature of the initial state? Could you be more specific?

≫ Lines 156: To create the 10 member ensembles within each set the individual simulations ran with different random perturbations in the stochastic physics, causing the atmospheric flow to diverge into different meteorological realizations.

Thank you for this valuable suggestion. The description about the stochastic physics has been added in the Methods section.

**Lines 161-168**

To create the 10 member ensembles within each set the individual simulations ran with different random perturbations in the stochastic physics, causing the atmospheric flow to diverge into different meteorological realizations. The stochastic physics introduces small random perturbations to the wind fields, temperature and moisture tendencies from some of the sub-grid parameterizations schemes including convection, gravity-wave drag, radiation and large-scale cloud microphysics. These perturbations are applied in a way that conserves energy, momentum and moisture but represents variability and uncertainty in unresolved physical processes, which has been shown to improve ensemble predictions on medium-range (Palmer et al., 2009; Tennant et al., 2011), seasonal (Weisheimer et al., 2011) and decadal timescales (Doblas-Reyes et al., 2009).

**8.** Lines 178-179, I would suggest the authors mention Fig. S1 here or even before to explain the definition of the two domains.

≫ Lines 178-179: As our study involves the sub-seasonal variations in monsoon onset and withdrawal, the monthly comparisons among South and East Asia precipitations from CPC and GPCP observations, ERA5 reanalysis and the UKESM1 simulations are shown in Fig. 1.

Thank you for this valuable suggestion. The description about Fig. S1 has been moved to the part of Model evaluation.

**Lines 212-214**

As our focus is primarily on the monsoon, here the model performance is evaluated by comparing against observations of the regional precipitation over South and East Asia. The division of South and East Asia (Fig. S1) used in this study follows Iturbide et al. (2020), which is adopted by IPCC (2021).

**9.** Line 215, missing a space between Cm and of.

≫ Lines 215: N2016 in Fig. 3(b) used All-India rainfall (AIR) to calculate the cumulative daily anomaly $C'_m(i)$ of AIR for day $i$ of year $m$:……

We feel sorry that we did not thoroughly check the format detail. The space has been added.

**Lines 196**

N2016 uses All-India rainfall (AIR) to calculate the cumulative pentad mean anomaly $C'_m(i)$ of AIR for pentad $i$ of year $m$:……

**10.** Lines, 272-274, the conclusion seems to be too general, and may not be the situation for both EASM and SASM. For example, how about the increase of SLP over China in Fig. 8c?

≫ Lines 272-274: The SCT reduction induced land warming yields a lower SLP anomaly over Asia continent compared with that over Indian and western Pacific oceans, which is favourable for the early/late transition of land-sea pressure difference in pre/post-monsoon season and a stronger SASM and EASM circulation in monsoon season.

≫Fig. 8:

[Figure]

Figure 8: Spatial distributions of the sea level pressure (unit: hPa) responses to the reductions in total aerosols (a, e and i), SCT aerosols (c, g and k) and ABS aerosols (d, h and l) over Asia during pre-monsoon (April-May; a-d), monsoon (June-August; e-h) and post-monsoon (September-October; i-l) seasons. Panels (b), (f) and (j) are the sum of the impacts of the reductions in the SCT and ABS. Black and pink dotted regions denote where the sea level pressure change is statistically significant at the 95% and 90% confidence level, respectively, according to a t-test.

Sorry, we didn't make it clear. According to your suggestions, (1) We have modified the Fig. 8 in order to show the changes in sea level pressure (SLP) caused by aerosol emission reductions more intuitively. Only the SLP changes with a confidence level of 95% or 90% according to the t-test are shown in the new Fig. 8 (Lines 1032-1038). The increase of SLP over China in original Fig. 8 (c) is insignificant and not shown in new Fig. 8 (c); (2) We have added a new figure (Fig. S3 in Supplement; Lines 48-56) to quantitatively examine the anomalous land-sea SLP difference between the Asian continent and its surrounding oceans and seas; (3) We have added more analysis and made a more careful statement about the impacts of SCT reduction on SLP changes over Asian continent and its adjacent oceans and seas in Lines 393-405.

**Lines 1032-1038 (new Figure 8)**

[Figure]

Figure 8: Spatial distributions of the sea level pressure (unit: hPa) responses to the reductions in total aerosols (a, e and i), SCT aerosols (c, g and k) and ABS aerosols (d, h and l) over Asia during pre-monsoon (April-May; a-d), monsoon (June-August; e-h) and post-monsoon (September-October; i-l) seasons. The sea level pressure responses are the difference between the aerosol-emission-perturbed and control runs. Panels (b), (f) and (j) are the sum of the impacts of the reductions in the SCT and ABS. Only the sea level pressure changes with a confidence level of 95% or 90% according to the t-test are shown.

**Lines 48-56 (Fig. S3; Supplement with tracked changes)**

[Figure]

**Figure S3. Time series of the anomalous land-sea sea level pressure (SLP) difference (unit: hPa) between the Asian continent part (including South Asia, East Asia, Tibet Plateau and East-Central Asia) adjacent to the ocean and its surrounding oceans and seas (including Northwest Pacific, tropical Indian Ocean, Bay of Bengal and Arabian Sea) to the reductions in total aerosols (b; gray line), SCT aerosols (a; red line) and ABS aerosols (a; blue line). The x-axis denotes the time (unit: pentad). The land-sea SLP difference responses are the difference between the aerosol-emission-perturbed and control runs. Purple line in Panel (b) represent the sum of the impacts of the reductions in the SCT and ABS. The shading area denote the standard deviation of the land-sea SLP difference anomaly. The sub-panel attached to Panel (a) gives the climatological land-sea SLP difference (unit: hPa) from control simulations. The region division used in this study refers to the sixth IPCC assessment report and is shown in Fig. S1.**

**Lines 393-405**

The SCT reduction induced land warming reduces the SLP over Asia continent but increase the SLP over Northwest

Pacific (Fig. 8c, g and k). The quantitative results of the anomalous land-sea SLP difference between the Asian conti- nent and its surrounding oceans and seas are shown in Fig. S3. The SCT reduction induces a negative land-sea SLP

difference anomaly throughout the year (Fig. S3 and Fig. 8c, g and k), which is favourable for the advance in the landsea SLP difference transition from positive to negative in spring and the delay in the transition from negative to positive in autumn. The negative anomalous land-sea SLP difference also leads to bigger land-sea SLP contrast and a stronger SASM and EASM circulation in the monsoon season. Note that the SLP changes in part of the oceanic areas adjacent to the SA region are consistent with the continental SLP changes, albeit with a smaller range of decrease (Fig. 8c, g and k). This could potentially be attributed to the reduced ACT that transported from Asian continent. However, the SLP decrease over these oceanic areas exerts negligible influence on the overall SCT-reduction-induced anomalous negative land-sea SLP difference between the Asian continent and adjacent oceans (Fig. S3).

**11.** Lines, 279-282, I would argue that Fig 8a seems to be more close to Fig 8c, indicating SCT dominating. Fig. 8i does not like Fig. 8j-l, indicating strong non-linearity?

≫ Lines 279-282: In addition, the anomalous land-sea SLP gradient between the Asian continent and the Indian and the western Pacific Oceans caused by the short-term total aerosols mitigation during monsoon season is dominated by the SCT aerosols and enhances the monsoon circulation over South and East Asia (Fig. 8e). In other seasons except summer, the land-sea SLP adjustments over Asia is controlled by the combined effects of SCT- and ABS-reductions (Fig. 8a and 8i).

Thank you for this valuable suggestion. We have added more analysis and made a more careful statement. We have added discussions about the Fig. 8 (a and i; see 10$^{th}$ Reply) and the dominant effects in regulating the SLP responses over Asian continent and its surrounding oceans and seas in Lines 411-421.

**Lines 411-421**

In addition, the anomalous land-sea SLP difference between the Asian continent and the topical Indian and Northwest Pacific Oceans caused by the short-term total aerosols mitigation during monsoon season is dominated by the SCT aerosols and enhances the monsoon circulation over South and East Asia (Fig. 8e). There is also a negative land-sea SLP difference anomaly due to the total aerosols mitigation in pre- and post-monsoon seasons (Fig. S3b), which is governed by the impacts of SCT-reduction. However, the spatial pattern of SLP adjustments during pre-monsoon season induced by total aerosol reduction shows a SLP increase over the seas of Southeast Asia, and both the impacts of SCT- and ABS-reduction (Fig. 8 c and d) contribute to the SLP increase over this region. Besides, the ABS-reduction has no significant impacts on the SLP adjustments during post-monsoon season (Fig. 8i). But the regions with signifi- cant SLP changes caused by total aerosol reduction are also inconsistent with those caused by the SCT reduction (Fig.

8i and k), indicating the strong non-linearity of atmospheric system.

**12.** Lines 295-300, should it be "atmospheric warming associated with the SCT reduction" according to Figure 7?

Should it be "The geopotential height increases in the uppermost troposphere …" instead of "The pressure ..."? I would argue that it is not so obvious in Figure 9 that the geopotential height changes strengthens the poleward pressure gradi- ent force in the north flank of this area.

≫ Lines 295-300: Coherent with the temperature perturbations shown in Fig. 7, both the geopotential height changes over South and East Asia during monsoon season caused by the emission reduction of total aerosols are dominated by the atmospheric cooling associated with the SCT reduction. The pressure increases in the uppermost troposphere (200-

500 hPa) north of 40°N over South and East Asia, which strengthens (weakens) the poleward pressure gradient force in the north (south) flank of this area.

Thank you for your correction. The relevant description has been corrected. Besides, the meridional gradient of geopo- tential height responses to aerosol reductions has also been added into Fig. 9 and Fig. S4 in order to intuitively display the changes in pressure gradient force.

**Lines 439-443**

Coherent with the temperature perturbations shown in Fig. 7, both the geopotential height changes over South and East

Asia during monsoon season caused by the emission reduction of total aerosols are dominated by the atmospheric warming associated with the SCT reduction. The geopotential height increases in the uppermost troposphere (200-500

hPa) around 40°N over South and East Asia, which strengthens (weakens) the poleward pressure gradient force in the north (south) flank of this area.

**Lines 1040-1047 (Figure 9)**

[Figure]

Figure 9: Zonal-mean geopotential height (unit: gpm) responses to the reductions in total aerosols (a and b), SCT aerosols (e and f), and ABS aerosols (g and h) during monsoon season over South Asia (70-90°E; a, c, e and g) and East Asia (100-120°E; b, d, f, h). Monsoon season is analyzed and based on the definitions from W2009 over South Asia and W2016 over East Asia. The geopotential height responses are the difference between the aerosol-emission-perturbed and control runs. Panels (c) and (d) are the sum of the impacts of the reductions in the SCT and ABS. Black lines denote the meridional gradient of GH response (unit: gpm m⁻¹; solid and dashed lines denote positive and negative values, respectively). Black and pink dotted regions denote where the geopotential height change is statistically significant at the 95% and 90% confidence level, respectively, according to a t-test.

Lines 60-67 (Figure S4; Supplement)

[Figure]

**Figure S4.** Zonal-mean geopotential height (unit: m) responses to the reductions in total aerosols (a and b), scattering (SCT) aerosols (e and f), and absorbing (ABS) aerosols (g and h) during monsoon season over South Asia (70-90°E; a, c, e and g) and East Asia (100-120°E; b, d, f, h). Monsoon season is analyzed and based on the definitions from N2016 over South Asia and G1983 over East Asia. Panels (c) and (d) are the sum of the impacts of the reductions in the SCT and ABS. Black lines denote the meridional gradient of GH response (unit: gpm m$^{-1}$; solid and dashed lines denote positive and negative values, respectively). Black and pink dotted regions denote where the geopotential height change is statistically significant at the 95% and 90% confidence level, respectively, according to a t-test.

**13.** Line 345, should be "may not be a linear summation of"?

≫ Lines 345: The SASM and EASM responses to the reductions in total aerosols may not a linear summation of the impacts of the reductions in individual aerosol type due to the nonlinearity of the atmospheric systems.

Sorry, this is a grammatical mistake. We have corrected the sentence.

**Lines 358-359**

However, the SASM and EASM responses to the reductions in total aerosols may not be a linear summation of the impacts of the reductions in individual aerosol type due to the nonlinearity of the atmospheric systems.

**14.** Lines 384-393, I would suggest the authors state the impacts for SCT and ABS in two separate sentences instead of putting antonyms in parentheses. It is easy to get lost when reading long sentences.

≫ Lines 384-393: The warming (cooling) induced by the SCT (ABS) reduction over South and East Asia during pre- and post-monsoon seasons favors early (late) transition of land-sea thermal contrast and SLP gradient in spring and late (early) transition in autumn, thus extending (shortening) the monsoon by advancing (delaying) its onset and delaying (advancing) the withdrawal. The change in pressure gradient force induced by SCT (ABS) aerosol reduction leads to an increase (decrease) in westerlies to the north of the upper-tropospheric jet center, leading to the northward (southward) displacement of the high-level easterly and westerly jet. The northward (southward) displacement of the high-level jet causes the anomalous moisture convergence (divergence) and upward (downward) motion at the lower level over north India and east China, eventually enhancing (weakening) the precipitation over South and East Asia during monsoon season. The stronger (weaker) SAH due to the land warming (cooling) induced by the reduction of SCT (ABS) also facilitates (hinders) the local convective development over northern South Asia and southern East Asia.

Thank you for this valuable suggestion. We have clarified the impacts of scattering (SCT) and absorbing (ABS) aerosols in two different sentences.

**Lines 532-543**

The warming induced by the SCT reduction over South and East Asia during pre- and post-monsoon seasons favors early transition of land-sea thermal contrast and SLP difference in spring and late transition in autumn, thus extending the monsoon by advancing its onset and delaying the withdrawal. The change in pressure gradient force induced by
SCT aerosol reduction leads to an increase in westerlies to the north of the upper-tropospheric jet center, leading to the
northward displacement of the high-level easterly and westerly jet. The northward displacement of the high-level jet
causes the anomalous moisture convergence and upward motion at the lower level over north India and east China,
eventually enhancing the precipitation over South and East Asia during monsoon season. The stronger SAH due to the
land warming induced by the reduction of SCT also facilitates the local convective development over northern South
Asia and southern East Asia. However, ABS reduction acts in the opposite sense in Asian climate responses, which
delays the transition of land-sea contrast in spring and advancing the transition in autumn, forces the Asian jet to move
southward, and weakens the SAH intensity.

**15.** Figure 2, I would suggest changing the title for middle column to be MON to be consistent with PRE and PST. Or
maybe it is better to just pre-monsoon, monsoon, and post-monsoon.

≫ Figure 2:

[Figure]

**Figure 2: Spatial distributions of the climatological mean (1985-2014) wind directions (vectors; unit: m s$^{-1}$) and wind speeds (shading; unit: m s$^{-1}$) at 200 hPa (a-i) and 850 hPa (j-r) from ERA5 reanalysis (a-c and j-l) and CMIP6-UKESM1 historical simulation (d-f and m-o) over Asia during pre-monsoon (April-May; a, d, g, j, m and p), monsoon (June-August; b, e, h, k, n and g) and post-monsoon (September-October; c, f, i, l, o and r) seasons. Panels (g-i) and (p-r) show the differences between the wind fields from the UKESM1 simulation and ERA5 reanalysis at 200 hPa and 850 hPa, respectively.**

Thank you for this valuable suggestion. We have unified the titles of the three columns in Fig. 2.

**Lines 981-986**

[Figure]

**Figure 2: Spatial distributions of the climatological mean (1985-2014) wind directions (vectors; unit: m s⁻¹) and wind speeds (shading; unit: m s⁻¹) at 200 hPa (a-i) and 850 hPa (j-r) from ERA5 reanalysis (a-c and j-l) and CMIP6-UKESM1 historical simulation (d-f and m-o) over Asia during pre-monsoon (April-May; a, d, g, j, m and p), monsoon (June-August; b, e, h, k, n and q) and post-monsoon (September-October; c, f, i, l, o and r) seasons. Panels (g-i) and (p-r) show the differences between the wind fields from the UKESM1 simulation and ERA5 reanalysis at 200 hPa and 850 hPa, respectively.**

**16.** Figure 3, could you change scale for left y-axis of panel b from day to pentad? It may be clearer to compare panel a and b. Is it possible to add values showing linearly combined SCT-75% and ABS-75%?

≫ Figure 3:

Figure 3: Box diagrams of the monsoon duration (red; unit: pentad for a, c and d, day for b) and precipitation (blue; unit: mm day$^{-1}$) over South Asia (a and b) and East Asia (c and d) in different simulations. Dots and middle horizontal lines inside boxes indicate mean and median values, respectively, and lower and upper sides of boxes indicate 25 and 75% range, respectively, and top and bottom line represent 5% and 95%, respectively. Panel (a) is derived based on the definition from Wang et al. (2009; hereafter referred to as W2009). Panel (b) is derived based on the definition from Noska and Misra (2016; hereafter referred to as N2016). Panel (c) is derived based on the definition from Wang, D. et al (2016; hereafter referred to as W2016). Panel (d) is derived based on the definition from Guo (1983; hereafter referred to as G1983).

Thank you for this valuable suggestion. The scale for left y-axis of Fig. 3(b) has been changed from day to pentad. The comparison between the Fig. 3(a) and (b), the comparison between the Fig. 3(c) and (d) and the relevant discussions have been added in Lines 271-286 and Lines 305-308. The linear addition of the impacts of the reductions in the SCT

and ABS has also been added in Fig. 3. The relevant descriptions about the linear addition of the impacts of reductions in the SCT and ABS are added in Lines 364-380.

**Lines 988-996 (Figure 3)**

[Figure]

**Figure 3: Box diagrams of the monsoon duration (red; unit: pentad) and precipitation (blue; unit: mm day⁻¹) over South Asia (a**
**and b) and East Asia (c and d) in different simulations. Dots and middle horizontal lines inside boxes indicate mean and median**
**values, respectively, and lower and upper sides of boxes indicate 25 and 75% range, respectively, and top and bottom line repre-**
**sent 5% and 95%, respectively. The boxes labelled SCT-75+ABS-75% in each panel are the linear addition of the impacts of the**

**reductions in the SCT and ABS.** Panel (a) is derived based on the definition from Wang et al. (2009; hereafter referred to as
**W2009).** Panel (b) is derived based on the definition from Noska and Misra (2016; hereafter referred to as N2016). Panel (c) is
**derived based on the definition from Wang, D. et al (2016; hereafter referred to as W2016).** Panel (d) is derived based on the defi-
**nition from Guo (1983; hereafter referred to as G1983).**

**Lines 271-286 (The comparison between the Fig. 3(a) and (b) and the relevant discussions. Note that the discus-**

**sions involved the monsoon onset and withdrawal adjustments (Fig. 4; shown in the 17[th] Reply))**

Note that using different definitions of monsoon onset/withdrawal dates may result in the variations in the monsoon duration response range although the SASM duration and precipitation adjustments in W2009 and N2016 are qualita- tively consistent. The SASM durations in N2016 from different simulation sets are basically 4-5 pentads longer than those in W2009 (Fig. 3a and b). The SCT-driven extension of the SASM duration based on N2016 (2 pentads) is also longer than that in W2009 (0.4 pentads). The difference in the SASM duration adjustments between W2009 and

N2016 can be attributed to the distinct selection of monsoon feature to characterize the monsoon subseasonal varia- tions. Syroka et al (2004) pointed out that the withdrawal of the SASM defined by the precipitation is much later than that defined by the monsoon circulation due to the late decrease in precipitation in southern India. The precipitation continues to increase in southern Indian after September associated with the winter monsoon (Bhanu Kumar et al.,

2004), while the SASM-related circulation characteristics becomes unclear in the meantime. Therefore, the SASM on- set dates based on N2016 is roughly the same with those based on W2019, but the withdrawal date is about 5 pentads later, resulting in the longer monsoon duration (Fig. 4a and b). Moreover, there exists an additional enhancement of monsoon precipitation over SA in the "SCT" set, which further leads to the later SASM withdrawal and longer SASM

duration in N2016 (Fig. 4b). Besides, the precipitation during early autumn is sensitive to the location and synop- tic/sub-synoptic systems (tropical cyclones, depressions, easterly waves, north-south trough activity and coastal con- vergence, etc; Bhanu Kumar et al., 2004), which possibly contributes to the larger variation range in the monsoon withdrawal date in N2016.

**Lines 305-308 (The comparison between the Fig. 3(c) and (d) and the relevant discussions. Note that the discus-**

**sions involved the monsoon onset and withdrawal adjustments (Fig. 4; shown in the 17[th] Reply))**

Compared to the EASM adjustments in W2016, the EASM show longer duration (about 3 pentads) in G1983 due to the later withdrawal (about 4 pentads). Zhu et al. (2012) has clarified that the climatological transition date of the zonal land-sea contrast in autumn over EASM-controlled region is about 3 pentads later than that of the monsoon circulation, which largely explained the relatively late monsoon withdrawal dates in G1983.

**Lines 364-380 (The description about the linear addition of the impacts of the reductions in the SCT and ABS. Note that the discussions involved the spatial pattern of SASM and EASM responses (Fig. 5 and 6; shown in the 1st Reply))**

Generally, the pattern of the anomalous precipitation and monsoon horizontal circulation over SA by adding the results of reducing SCT and ABS aerosols are similar to the results of reducing total aerosols, especially for the W2009 (Fig. 5). However, the precipitation north of 30°N shows a reduction in the validity of the linear addition assumption compared to the precipitation change in the simulation of reducing total aerosols, although most of the reduced precipitation does not pass the significance test ($p < 0.05$). There's also significantly increased precipitation over the southern part of SA in the linear addition, which is contributed by the impacts of SCT reduction. Additionally, an easterly anomaly appears over the Arabian Sea (10-20°N) in N2016 (Fig. 5f) as the linear addition of Fig. 5g and 5h, while an SASM westerly flow is enhanced over this region in the simulations of reducing total aerosols (Fig. 5e). The dominated impacts of SCT reduction and non-linear effects between the SCT and ABS contribute to the enhanced SASM westerly in Fig. 5e. The general feature of precipitation and circulation responses over the EA continent (north of 15°N) in the linear addition are also consistent with that in the simulation of simultaneous SCT and ABS reductions (Fig. 6), except for the insignificant decreased precipitation contributed by the impacts of ABS reduction (Fig. 6b and 6f). For the quantitative results of regional precipitation adjustments, the linear addition results show an increased precipitation in both SA and EA compared with the CTRL results (Fig. 3). The increased precipitation amount is less than the results of reducing total aerosols due to the simple addition of precipitation change caused by ABS reduction. However, the results of linear addition are inconsistent with the total aerosol reduction results in terms of the SASM and EASM duration variations, indicating that the impacts of reducing SCT or ABS alone on monsoon subseasonal variability cannot be simply added up.

**17.** Figure 6, similar as Figure 3, could you change scale of panel b from day to pentad for better comparison and consistency?

≫ Figure 6:

[Figure]

Figure 6: Same as Figure 3, but for the monsoon onset dates (yellow; unit: pentad for a, c and d, day for b), withdrawal dates (green; unit:
pentad for a, c and d, day for b) and duration (red; unit: pentad for a, c and d, day for b).

Thank you for this valuable suggestion. In the revised manuscript, the serial number of Fig. 6 is changed to Fig. 4. The
scale for left y-axis of Fig. 4(b) has been changed from day to pentad. The linear addition of the impacts of the reduc-
tions in the SCT and ABS has also been added in Fig. 4. **The comparison between the Fig. 4(a) and (b), the com-**

**parison between the Fig. 4(c) and (d) and the relevant discussions have been added in Lines 271-286 and Lines**

**305-308, which can be seen in the 16th Reply.**

**Lines 999-1002 (Fig. 4)**

[Figure]

**Figure 4: Same as Figure 3, but for the monsoon onset dates (yellow; unit: pentad), withdrawal dates (green; unit: pentad) and**
**duration (red; unit: pentad).** The yellow and green dashed lines denote the mean values of monsoon onset and withdrawal in the
CTRL simulation set, respectively.